# `G-Memory`: Tracing Hierarchical Memory for Multi-Agent Systems

**Guibin Zhang**[*1], **Muxin Fu**[*2], **Kun Wang**[3†], **Guancheng Wan**[4], **Miao Yu**[5], **Shuicheng Yan**[1†]

[1]NUS, [2]Tongji University, [3]NTU, [4]WHU, [5]A*STAR

[*] Equal Contribution, [†] Corresponding author

✉ wang.kun@ntu.edu.sg, yansc@comp.nus.edu.sg

## Abstract

Large language model (LLM)-powered multi-agent systems (MAS) have demonstrated cognitive and execution capabilities that far exceed those of single LLM agents, yet their capacity for self-evolution remains hampered by underdeveloped memory architectures. Upon close inspection, we are alarmed to discover that prevailing MAS memory mechanisms (1) are overly simplistic, completely disregarding the nuanced inter-agent collaboration trajectories, and (2) lack cross-trial and agent-specific customization, in stark contrast to the expressive memory developed for single agents. To bridge this gap, we introduce `G-Memory`, a hierarchical, agentic memory system for MAS inspired by organizational memory theory [1], which manages the lengthy MAS interaction via a three-tier graph hierarchy: insight, query, and interaction graphs. Upon receiving a new user query, `G-Memory` performs bi-directional memory traversal to retrieve both *high-level, generalizable insights* that enable the system to leverage cross-trial knowledge, and *fine-grained, condensed interaction trajectories* that compactly encode prior collaboration experiences. Upon task execution, the entire hierarchy evolves by assimilating new collaborative trajectories, nurturing the progressive evolution of agent teams. Extensive experiments across five benchmarks, three LLM backbones, and three popular MAS frameworks demonstrate that `G-Memory` **improves success rates in embodied action and accuracy in knowledge QA by up to** $20.89\%$ **and** $10.12\%$**,** respectively, without any modifications to the original frameworks. Our codes are available at https://github.com/bingreeky/GMemory.

## 1  Introduction

As Large Language Models (LLMs) continue to redefine the frontier of artificial intelligence, *LLM-driven agents* have exhibited unprecedented prowess in perception [2, 3, 4, 5], planning [6, 7, 8], reasoning [9, 10], and action [11, 12], which have catalyzed remarkable progress across diverse downstream domains, including code generation [13, 14], data analysis [15], embodied tasks [16] and autonomous driving [3, 17, 18]. Building upon the impressive competencies of single agents, LLM-based Multi-Agent Systems (MAS) have been demonstrated to push the boundaries of single model capacity [19, 20, 21]. Similar to collective intelligence arising from human social collaboration [22, 23, 24], MAS orchestrates multiple agents [25, 26, 27], whether through cooperation [28, 29, 30, 31] or competition [32, 33, 34], to transcend the cognitive and specialized limitations of solitary agents.

**Self-Evolving Agents.**  What especially characterizes LLM agents is their *self-evolving capacity*, *i.e.*, the ability to continuously adapt and improve through interactions with the environment, as seen in prior works where such adaptability has led to two- to three-fold quantitative improvements [35]. The central driving force behind such self-evolving nature is **memory mechanism** of agents [36, 37], which parallels human abilities to accumulate knowledge, process past experiences, and retrieve

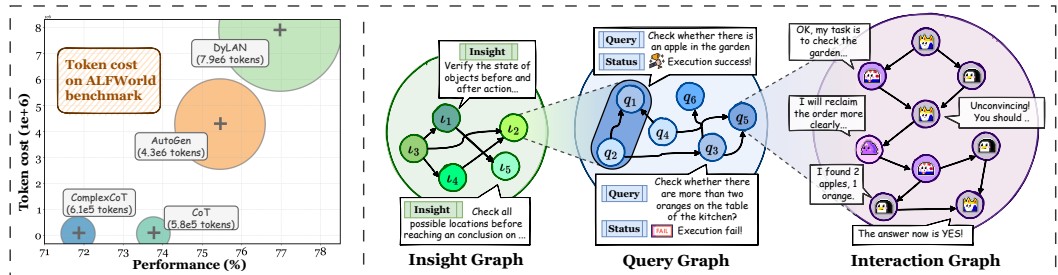

Figure 1: (*Left*) We report the token cost of several single-agent and MAS baselines on ALFWorld benchmark; (*Right*) The overview of `G-Memory`'s three-tier hierarchical memory architecture, encompassing the insight graph, query graph and interaction (utterance) graph.

relevant information. Previous successful memory mechanism designs, including both inside-trial memory (*i.e.*, context retained within solving one single query) and cross-trial memory (*i.e.*, experience accumulated across multiple tasks) [38], have empowered agents to excel in diverse applications such as personalized chat [36, 39, 40], recommendation [41], embodied action [42, 16], and social simulation [19, 43, 44], enabling them to evolve into experiential learners that effectively leverage past experiences and world knowledge.

**Self-Evolving MAS.** However, such self-evolving capacity remains largely absent in multi-agent systems. Most existing MAS are still constrained by manually defined workflows, such as the Standard Operating Procedures (SOP) in MetaGPT [21] and ChatDev [45], or rely on pre-defined communication topologies in MacNet [46] and AgentPrune [30]. More recent automated MASs, such as GPTSwarm [47], ADAS [48], AFlow [49], and MaAS [50] have made it to automatically optimize inter-agent topologies or prompts, which, nevertheless, ultimately yield giant and cumbersome MAS architectures, lacking the agility to self-adjust with accumulated collaboration experience.

**Memory for MAS.** The absence of the aforementioned self-evolving capacity is, in fact, rooted in the lack of memory mechanisms specifically tailored for MAS. One may challenge this claim from two perspectives: ❶ *Do existing MASs lack memory mechanisms altogether?* Not entirely. Classical MAS frameworks such as MetaGPT, ChatDev, and Exchange-of-Thought [51] incorporate memory-related designs. However, these are often limited to inside-trial memory [51], while cross-trial memory, if present, remains rudimentary—typically involving the transmission of overly condensed artifacts (*e.g.*, final solutions or execution results) [21, 45, 46], and failing to enable meaningful learning from collaborative experience. ❷ *Why not directly transfer existing single-agent memory mechanisms to MAS?* Unfortunately, such a transfer is far from straightforward. The inherent nature of MAS, *i.e.*, multi-turn orchestration across multiple agents [26, 27], leads to substantially longer task-solving trajectories compared to single-agent settings (up to $10\times$ more tokens, as demonstrated by Figure 1 (*Left*)). This poses a significant challenge to traditional retrieval-based memory designs [36, 37, 16], as naive feeding of the entire long-context trajectory without proper abstraction from a collaborative perspective offers little benefit. Given the aforementioned challenges, a natural question arises:

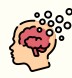 *How can we design a memory mechanism capable of storing, retrieving, and managing the lengthy interaction history of multi-agent systems, such that agent teams can benefit from concise and instructive experience and insights?*

**The Present Work: `G-Memory`.** In response to the above question, we introduce a *Graph-based Agentic Memory Mechanism for LLM-based Multi-Agent Systems*, dubbed `G-Memory`, which manages the complex and lengthy interaction history of MAS through a three-tier hierarchical graph structure:

- ✳ **Insight Graph**, which abstracts generalizable insights from historical experience;
- ✳ **Query Graph**, which encodes meta-information of task queries and their connectivity;
- ✳ **Interaction Graph**, which stores fine-grained textual communication logs among agents.

Figure 1 (*Right*) visualizes these structures, and their formal definitions are placed in Section 3. When a new query arrives, `G-Memory` efficiently retrieves relevant query records by leveraging the topology of the query graph, and then traverses *upward* (*i.e.*, query→insight graph) to extract associated high-level insights and *downward* (*i.e.*, query→interaction graph) to identify core interaction subgraphs that are most pertinent to the task at hand, thereby mitigating information overload. Based on the

retrieved memory, `G-Memory` offers actionable guidance to the MAS, *e.g.*, division of labor, task decomposition, and lessons from past failures. Upon the completion of a task, all three levels of the memory hierarchy are updated in an agentic manner, with newly distilled insights, enriched query records, detailed MAS trajectories, and their level of detailed associations. Through this refinement, `G-Memory` functions as a plug-and-play module that can be seamlessly embedded into mainstream MAS frameworks, empowering evolving inter-agent collaboration and collective intelligence.

Our contributions are summarized as follows:

❶ **Bottleneck Identification.** We conduct a thorough review of existing multi-agent systems and identify a fundamental bottleneck in their self-evolving capabilities, which is largely attributed to the oversimplified memory architectures.

❷ **Practical Solution.** We propose `G-Memory`, a hierarchical agentic memory architecture for MAS, which models complex and prolonged inter-agent collaboration through a three-tier structure comprising insight, query, and interaction graphs.

❸ **Experimental Evaluation.** Extensive experiments across five benchmarks show that `G-Memory` is (**I**) *high-performing*, improving state-of-the-art MAS by up to 20.89% and 10.12% on embodied action and knowledge QA tasks, respectively; and (**II**) *resource-friendly*, maintaining comparable or even lower token usage than mainstream memory designs.

## 2 Related Works

**Single-Agent Memory.** Memory serves as a primary driving force for agents to accumulate experiences and explore the world through interactions with the environment [52, 53, 54, 55]. It plays a critical role in both *task-solving* and *social simulation* LLM agents, and this work primarily focuses on the former. Early research on agent memory was confined to simple inside-trial memory, mainly addressing limitations posed by the LLM context window in chatbot applications, including MemoryBank [36], ChatDB [39], MemoChat [40], and MemGPT [37], which typically adopt retrieval-augmented generation (RAG)-style, similarity-based chunk retrieval. Subsequent developments have progressed toward more cognitively inspired memory architectures, including (1) memory scope extended to cross-trial memory like ExpeL [42] and Synapse [56]; (2) application domains broadened to include computer control [56], embodied action [57], scientific discovery [58], coding and reasoning [59]; and (3) management techniques evolved from coarse-grained textual similarity toward more sophisticated abstraction and summarization of acquired knowledge and experiences [19], as seen in A-Mem [60], Mem0 [61] and MemInsight [62]. More discussions are in Appendix D.

**Memory in Multi-agent System.** However, the memory mechanisms tailored for MAS remain markedly underexplored. Some representative frameworks, such as LLM-Debate [20, 33] and Mixture-of-Agent [63], omit memory components altogether. Others merely adopt simplistic inside-trial memory schemes [46, 51]. Even in frameworks that attempt cross-trial memory [45], the memory is merely compressed as the final outcome artifacts, overlooking the nuanced agent interactions. Collectively, there is a pressing need for a principled memory architecture that can capture, organize, and retrieve the inherently intricate task-solving processes unique to MAS [38].

**LLM-based Multi-Agent Systems.** Our work focuses on *task-solving* MAS, which, unlike their single-agent counterparts, often lack the capacity for continual evolution through interaction with the environment [64, 65]. Early frameworks such as AutoGen [13], CAMEL [24], and AgentVerse [66] rely entirely on pre-defined workflows. More recent efforts [67, 68, 49, 48, 69, 31] introduce a degree of adaptivity by generating dynamic MAS in response to environmental feedback. However, such evolution is often *one-shot*: for example, AFlow [49] employs Monte Carlo Tree Search to construct a complex MAS tailored to a specific task domain, which yet lacks the capacity to evolve with increasing task exposure or transfer across domains [50, 70]. From this perspective, constructing MAS with genuine self-evolving capabilities remains an open and challenging research frontier.

## 3 Preliminary

In this section, we establish the notation and formalize key concepts of multi-agent systems and `G-Memory`'s hierarchical memory architecture.

**Multi-agent System Formalization.** Consider a multi-agent framework represented by a directed graph $\mathcal{G} = (\mathcal{V}, \mathcal{E})$, where $|\mathcal{V}| = N$ is the number of agents and $\mathcal{E} \subseteq \mathcal{V} \times \mathcal{V}$ defines their communication

channels. Each node $C_i \in \mathcal{V}$ corresponds to an individual agent described by the quadruple:

$$C_i = (\mathsf{Base}_i, \mathsf{Role}_i, \mathsf{Mem}_i, \mathsf{Plugin}_i), \tag{1}$$

where $\mathsf{Base}_i$ denotes the underlying large language model instance, $\mathsf{Role}_i$ specifies the agent's designated role or persona, $\mathsf{Mem}_i$ encapsulates its memory state, including past interactions or external knowledge stores, and $\mathsf{Plugin}_i$ is the set of auxiliary tools (*e.g.*, web-search engine).

Upon receiving a user query $Q$, the system evolves through $T$ synchronous communication epochs. At each epoch $t$, we derive a topological ordering $\pi = [\pi_1, \ldots, \pi_N]$ of the nodes such that if there is an edge from $\pi_j$ to $\pi_k$, then $j < k$, which guarantees that every agent processes its inputs only after all its predecessors have acted. For each agent $C_i$ in $\pi$, its output at iteration $t$ is computed as:

$$r_i^{(t)} = C_i\Big(P_{\mathsf{sys}}^{(t)}, Q, \{r_j^{(t)} : C_j \in \mathcal{N}^-(C_i)\}\Big),$$

where: $r_i^{(t)}$ denotes the response generated by $C_i$ (which may include reasoning steps, intermediate analyses, or final proposals), $P_{\mathsf{sys}}^{(t)}$ comprises global instructions (including each agent's $\mathcal{R}_i$), $\mathcal{N}^-(C_i)$ is the set of in-neighbors of $C_i$, whose outputs serve as contextual inputs. After all agents have acted, a global aggregation operator $\mathcal{A}$ fuses the collection of responses into an interim solution $a^{(t)}$:

$$a^{(t)} = \mathcal{A}(r_1^{(t)}, \ldots, r_N^{(t)}).$$

Common implementations for $\mathcal{A}$ include majority voting schemes [47], hierarchical summarization via dedicated aggregator agents [13, 30], or simply adopting the final agent's output as the answer [46]. These epochs iterate for $t = \{1, \ldots, T\}$ until either a preset limit is reached or an early-stopping criterion is met [71], producing the final response $a^{(T)}$ to the query $Q$.

**Memory Architecture.** Our proposed `G-Memory` orchestrates and manages the memory of multi-agent systems via the following three hierarchical graph structures:

[✱] **Interaction Graph (Utterance Graph).** For query $Q$, let $\mathcal{G}_{\mathsf{inter}}^{(Q)} = (\mathcal{U}^{(Q)}, \mathcal{E}_{\mathsf{u}}^{(Q)})$ denote its interaction trajectory, where (i) nodes $\mathcal{U}^{(Q)} = \{u_i\}$ represent atomic utterances, with each $u_i \triangleq (\mathcal{A}_i, m_i)$ containing $\mathcal{A}_i \in \mathcal{V}$ (speaking agent), and $m_i$ (textual content), (ii) Edges $\mathcal{E}_{\mathsf{u}}^{(Q)} \subseteq \mathcal{U}^{(Q)} \times \mathcal{U}^{(Q)}$ follow temporal relationships: $(u_j, u_k) \in \mathcal{E}_{\mathsf{u}}^{(Q)} \iff u_j$ is transmitted to and inspires $u_k$.

[✱] **Query Graph.** The query graph, storing previously tackled queries and metadata, is as follows:

$$\mathcal{G}_{\mathsf{query}} = (\mathcal{Q}, \mathcal{E}_{\mathsf{q}}) = \left( \{Q_i, \Psi_i, \mathcal{G}_{\mathsf{inter}}^{(Q_i)}\}_{i=1}^{|\mathcal{Q}|}, \mathcal{E}_{\mathsf{q}} \right), \tag{2}$$

where $\mathcal{Q} = \{q_i\}$ is the node set, node $q_i \triangleq (Q_i, \Psi_i, \mathcal{G}_{\mathsf{inter}}^{(Q_i)})$ is composed of the original query $Q_i$, task status $\Psi_i \in \{\mathsf{Failed}, \mathsf{Resolved}\}$, and its associated interaction graph $\mathcal{G}_{\mathsf{inter}}^{(Q_i)}$. The edges $\mathcal{E}_{\mathsf{q}} \subseteq \mathcal{Q} \times \mathcal{Q}$ encode semantic relationships between queries. The query graph enables retrieval beyond coarse metrics such as embedding similarity, with its meticulous topology.

[✱] **Insight Graph.** The highest-level insight graph is featured as follows:

$$\mathcal{G}_{\mathsf{insight}} = (\mathcal{I}, \mathcal{E}_{\mathsf{i}}) = \left( \underbrace{\langle \kappa_k, \Omega_k \rangle}_{\iota_k}{}_{k=1}^{|\mathcal{I}|}, \mathcal{E}_{\mathsf{i}} \right), \tag{3}$$

where the node set $\mathcal{I} = \{\iota_k\}$ represents distilled insights, each node $\iota_k$ is composed of the insight content $\kappa_k$ and the set of supporting queries $\Omega_k \subseteq \mathcal{Q}$. The edges $\mathcal{E}_{\mathsf{i}} \subseteq \mathcal{I} \times \mathcal{I} \times \mathcal{Q}$ forming hyper-connections where $(\iota_m, \iota_n, q_j)$ indicates insight $\iota_m$ contextualizes $\iota_n$ through query $q_j$.

## 4 G-Memory

This section outlines the management workflow of `G-Memory`, as illustrated in Figure 2. Specifically, upon the arrival of a new query $Q$, `G-Memory` first conducts coarse-grained retrieval to identify pertinent trajectory records (▷ Section 4.1). It then performs bi-directional hierarchical memory traversal: upward to retrieve collective cognitive insights, and downward to distill concrete procedural trajectories (▷ Section 4.2). After the memory-augmented MAS completes the query execution, the hierarchical memory architecture is jointly updated based on environmental feedback, thereby achieving the institutionalization of group knowledge (▷ Section 4.3).

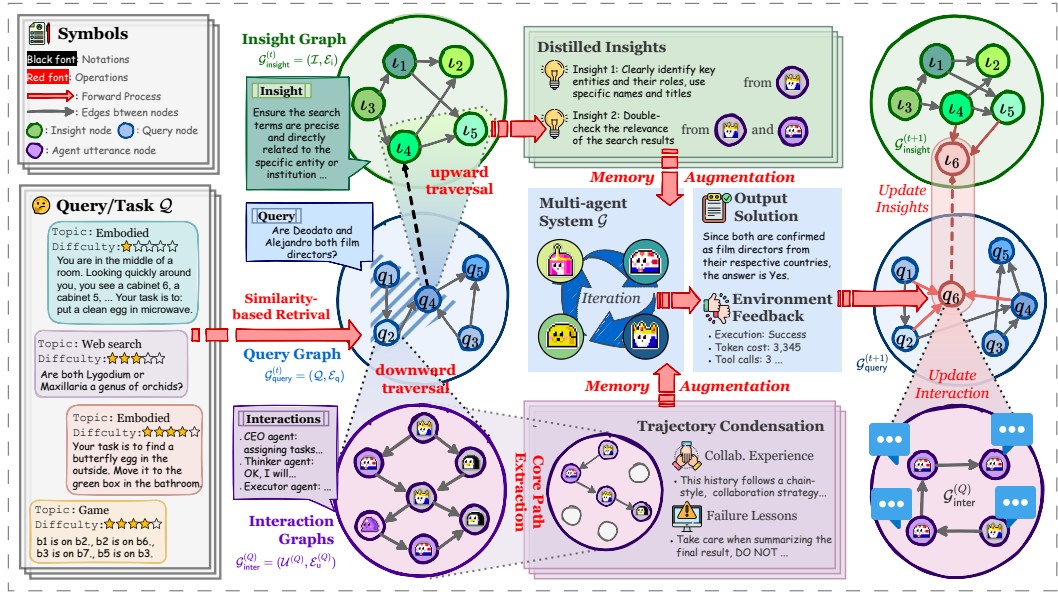

Figure 2: The overview of our proposed G-Memory.

## 4.1 Coarse-grained Memory Retrieval

As a plug-in designed for seamless integration into mainstream MAS, G-Memory is triggered when the MAS $\mathcal{G}$ encounters a new user query $Q$. As emphasized in organizational memory theory [1], efficient knowledge retrieval typically begins with broadly relevant schemas prior to more fine-grained access. Following this principle, G-Memory first performs a coarse-grained similarity-based retrieval over the query graph $\mathcal{G}_{\text{query}}$ to efficiently obtain a sketched set of queries $\mathcal{Q}^{\mathcal{S}}$:

$$\mathcal{Q}^{\mathcal{S}} = \underset{q_i \in \mathcal{Q} \text{ s.t. } |\mathcal{Q}^{\mathcal{S}}|=k}{\arg \text{top-k}} \left( \frac{\mathbf{v}(Q) \cdot \mathbf{v}(q_i)}{|\mathbf{v}(Q)| \, |\mathbf{v}(q_i)|} \right), \tag{4}$$

where $\mathbf{v}(\cdot)$ maps queries into fixed-length embeddings using models such as MiniLM [72]. While Equation (4) retrieves semantically similar historical queries, the similarity may be only superficial or noisy. Therefore, G-Memory further enlarges the relevant set via **hop expansion** on the query graph:

$$\tilde{\mathcal{Q}}^{\mathcal{S}} = \mathcal{Q}^{\mathcal{S}} \cup \left\{ Q_k \in \mathcal{Q} \mid \exists Q_j \in \mathcal{Q}^{\mathcal{S}}, \, Q_k \in \mathcal{N}^+(Q_j) \cup \mathcal{N}^-(Q_j) \right\}, \tag{5}$$

where $\tilde{\mathcal{Q}}^{\mathcal{S}}$ is augmented with the 1-hop neighbors of $\mathcal{Q}^{\mathcal{S}}$ on the query graph $\mathcal{G}_{\text{query}}$, and $\mathcal{N}^+(\cdot)$ and $\mathcal{N}^-(\cdot)$ denote the out-neighborhood and in-neighborhood of node $Q_j$, respectively. However, it is suboptimal to directly feed these relevant records as input akin to certain single-agent memory systems [40, 37]. On one hand, the excessive context length may overwhelm the LLM; on the other hand, agents in MAS play distinct roles and should be assigned *specialized* memory tailored to their functions. To address this, the next section introduces a bi-directional processing scheme in G-Memory that operates over both abstract and fine-grained memory levels.

## 4.2 Bi-directional Memory Traversal

Subsequent to identifying the expanded set of relevant query nodes $\tilde{\mathcal{Q}}^{\mathcal{S}}$ within $\mathcal{G}_{\text{query}}$, G-Memory executes a **bi-directional memory traversal** to furnish multi-granularity memory support. Specifically, G-Memory first performs an *upward traversal* ($\mathcal{G}_{\text{query}} \rightarrow \mathcal{G}_{\text{insight}}$), retrieving insight nodes that may provide high-level guidance for the current task:

$$\mathcal{I}^{\mathcal{S}} = \Pi_{\mathcal{Q} \rightarrow \mathcal{I}}(\tilde{\mathcal{Q}}^{\mathcal{S}}), \, \Pi_{\mathcal{Q} \rightarrow \mathcal{I}}(\mathcal{S}_q) \triangleq \{\iota_k \in \mathcal{I} \mid \Omega_k \cap \mathcal{S}_q \neq \emptyset\}, \tag{6}$$

where $\Pi_{\mathcal{Q} \rightarrow \mathcal{I}}$ is a query-to-insight projector that identifies all the insight nodes whose supporting query sets intersect with the input query set, and the retrieved insights $\mathcal{I}^{\mathcal{S}}$ encapsulate distilled, generalized knowledge potentially relevant for orienting the MAS $\mathcal{G}$'s strategic approach to $Q$.

Beyond generalized insights, the fine-grained textual interaction history of the MAS is equally valuable, as it reveals the underlying reasoning patterns that led to successful or failed collaborations [67, 73, 74]. To utilize these concisely, in the downward traversal ($\mathcal{G}_{\text{query}} \rightarrow \mathcal{G}_{\text{interaction}}$),

`G-Memory` employs an LLM-facilitated graph sparsifier $\mathcal{S}_{\text{LLM}}(\cdot, \cdot)$ to extract the core subgraph that encapsulates essential inter-agent collaboration:

$$\{\hat{\mathcal{G}}_{\text{inter}}^{Q_i}\}_{i=1}^{|M|} = \Big\{ \mathcal{S}_{\text{LLM}}(\mathcal{G}_{\text{inter}}^{(Q_j)}, Q) \mid q_j \in \underset{\{q'_k \in \tilde{\mathcal{Q}}^{\mathcal{S}}\} \text{ s.t. } |\cdot|=M}{\text{argtop-M}} \mathcal{R}_{\text{LLM}}(Q, q'_k) \Big\}, \tag{7}$$

where $\mathcal{R}_{\text{LLM}}(Q, q_j)$ rates the relevancy of historical queries w.r.t. $Q$, and the sparsifier $\mathcal{S}_{\text{LLM}}(\mathcal{G}_{\text{inter}}^{(Q_j)}, Q)$ constructs a sparsified graph $\hat{\mathcal{G}}_{\text{inter}}^{(Q_j)} = (\hat{\mathcal{U}}^{(Q_j)}, \hat{\mathcal{E}}_{\text{u}}^{(Q_j)})$ from the original $\mathcal{G}_{\text{inter}}^{(Q_j)}$ by identifying and retaining dialogue elements. Please refer to Appendix C for their implementations.

Upon completing the bi-directional traversal, we obtain both generalizable insights ($\mathcal{I}^{\mathcal{S}}$) and detailed collaborative trajectories ($\{\hat{\mathcal{G}}_{\text{inter}}^{Q_i}\}_{i=1}^{|M|}$). `G-Memory` then proceeds to provide specialized memory support for each agent $\mathcal{C} \in \mathcal{V}$ within the MAS $\mathcal{G}$.

$$\text{Mem}_i \leftarrow \Phi\left( \mathcal{I}^{\mathcal{S}}, \{\hat{\mathcal{G}}_{\text{inter}}^{Q_i}\}_{i=1}^{|M|}; \text{Role}_i, Q \right), \ \forall C_i = (\text{Base}_i, \text{Role}_i, \text{Mem}_i, \text{Plugin}_i) \in \mathcal{V}, \tag{8}$$

where the operator $\Phi(\cdot; \cdot)$ evaluates the utility and relevance of each insight $\iota_k \in \mathcal{I}^{\mathcal{S}}$ and sparsified interaction graph $\hat{\mathcal{G}}_{\text{inter}}^{(Q_j)}$ concerning the agent's specific role $\text{Role}_i$ and the task $Q$ (see Appendix C). Based on this evaluation, $\Phi$ intializes each agent's internal memory state $\text{Mem}_i$ with filtered insights, interaction snippets, summaries thereof, equipping it with pertinent historical context before it participates in the subsequent reasoning epochs of the MAS. It is worth noting that `G-Memory` is invoked at the onset of solving query $Q$ in our implementation. However, practitioners may flexibly configure more fine-grained invocation strategies, such as at the beginning of each MAS dialogue round or selectively for specific agents, based on their needs.

### 4.3 Hierarchy Memory Update

After completing memory augmentation for each agent, the system $\mathcal{G}$ is executed as outlined in Section 3, yielding a final solution $a^{(T)}$ and receiving environmental feedback, including execution status $\Psi_i \in \{\text{Failed}, \text{Resolved}\}$, token usage, and other performance metrics. Subsequently, `G-Memory` updates its hierarchical memory architecture to incorporate this new query. At the **interaction level**, `G-Memory` traces each agent's utterances to construct the interaction graph $\mathcal{G}_{\text{inter}}^{(Q)}$, which is then stored. At the **query level**, a new query node is instantiated and added to the query graph $\mathcal{Q}_{\text{query}}$:

$$q_{\text{new}} \leftarrow (Q, \Psi, \mathcal{G}_{\text{inter}}^{(Q)}), \ \mathcal{N}_{\text{conn}} \leftarrow \mathcal{Q}^{\mathcal{R}} \cup \Big( \bigcup_{\iota_k \in \mathcal{I}^{\mathcal{S}}} \Omega_k \Big),$$
$$\mathcal{E}_{\text{new}} \leftarrow \{(q_n, q_{\text{new}}) \mid q_n \in \mathcal{N}_{\text{conn}}\}, \ \mathcal{G}_{\text{query}}^{\text{next}} \leftarrow (\mathcal{Q} \cup \{q_{\text{new}}\}, \mathcal{E}_{\text{q}} \cup \mathcal{E}_{\text{new}}), \tag{9}$$

where edges are established between $q_{\text{new}}$ and (ii) the set $\mathcal{Q}^{\mathcal{R}}$ containing the top-$M$ relevant historical queries identified in Equation (7), and (ii) the set of queries $\bigcup_{\iota_k \in \mathcal{I}_{\text{ret}}} \Omega_k$ that support the insights $\mathcal{I}^{\mathcal{S}}$ utilized for solving $Q$. $\mathcal{G}_{\text{query}}^{\text{next}}$ denotes the updated query graph.

Finally, at the **insight level**, `G-Memory` integrates the learning from the completed query $Q$ into the insight graph $\mathcal{G}_{\text{insight}} = (\mathcal{I}, \mathcal{E}_{\text{i}})$. First, possible new insights summarizing the experience are generated and structurally linked via a summarization function $\mathcal{J}(\cdot, \cdot)$ (see prompt in Appendix C) as follows:

$$\iota_{\text{new}} = (\mathcal{J}(\mathcal{G}_{\text{inter}}^{(Q)}, \Psi), \{q_{\text{new}}\}), \ \mathcal{E}_{\text{i, new}} \leftarrow \{(\iota_k, \iota_{\text{new}}, q_{\text{new}}) \mid \iota_k \in \mathcal{I}^{\mathcal{S}}\}$$
$$\mathcal{G}_{\text{insight}}' \leftarrow (\mathcal{I} \cup \{\iota_{\text{new}}\}, \mathcal{E}_{\text{i}} \cup \mathcal{E}_{\text{i, new}}) \tag{10}$$

where edges are added to connect the previously utilized insights which inspires the completion of $Q$ in Equation (6). Afterward, the supporting query sets ($\Omega_k$) for the utilized insights ($\mathcal{I}^{\mathcal{S}}$) are updated to include $q_{\text{new}}$, reflecting their relevance to this successful (or failed) application:

$$\mathcal{I}^{\text{next}} \leftarrow (\mathcal{I} \setminus \mathcal{I}_{\text{ret}}) \cup \{(\kappa_k, \Omega_k \cup \{q_{\text{new}}\}) \mid \iota_k = (\kappa_k, \Omega_k) \in \mathcal{I}_{\text{ret}}\} \cup \{\iota_{\text{new}}\}$$
$$\mathcal{G}_{\text{insight}}^{\text{next}} \leftarrow (\mathcal{I}^{\text{next}}, \mathcal{E}_{\text{i}} \cup \mathcal{E}_{\text{i, new}}), \tag{11}$$

where the final node set $\mathcal{I}^{\text{next}}$ incorporates the new insight and the updated versions of the utilized insights, and the resulting graph $\mathcal{G}_{\text{insight}}^{\text{next}}$ thus encapsulates the integrated knowledge. This continuous update cycle across all hierarchical levels enables `G-Memory` to learn and adaptively refine its collective memory based on ongoing experience.

Table 1: Performance comparison with single/multi-agent memory architectures on five benchmarks. The underlying LLM backbone is `GPT-4o-mini`. We highlight the best and second best results.

| MAS | Memory | ALFWorld | SciWorld | PDDL | HotpotQA | FEVER | Avg. |
|---|---|---|---|---|---|---|---|
| AutoGen 
 COLM 2024 | No-memory | $77.61_{\uparrow 0.00}$ | $54.49_{\uparrow 0.00}$ | $23.53_{\uparrow 0.00}$ | $28.57_{\uparrow 0.00}$ | $57.13_{\uparrow 0.00}$ | $48.27_{\uparrow 0.00}$ |
| | Voyager | $85.07_{\uparrow 7.46}$ | $62.36_{\uparrow 7.87}$ | $24.56_{\uparrow 1.03}$ | $32.32_{\uparrow 3.75}$ | $63.27_{\uparrow 6.14}$ | $53.52_{\uparrow 5.25}$ |
| | MemoryBank | $74.96_{\downarrow 2.65}$ | $53.11_{\downarrow 1.38}$ | $20.41_{\downarrow 3.12}$ | $33.67_{\uparrow 5.10}$ | $61.22_{\uparrow 4.09}$ | $48.67_{\uparrow 0.40}$ |
| | Generative | $86.36_{\uparrow 8.75}$ | $61.19_{\uparrow 6.70}$ | $25.53_{\uparrow 2.00}$ | $31.63_{\uparrow 3.06}$ | $60.20_{\uparrow 3.07}$ | $52.98_{\uparrow 4.71}$ |
| | MetaGPT | $81.34_{\uparrow 3.73}$ | $61.91_{\uparrow 7.42}$ | $21.63_{\downarrow 1.90}$ | $32.67_{\uparrow 4.10}$ | $62.67_{\uparrow 5.54}$ | $52.04_{\uparrow 3.77}$ |
| | ChatDev | $79.85_{\uparrow 2.24}$ | $50.96_{\downarrow 3.53}$ | $16.65_{\downarrow 6.88}$ | $24.49_{\downarrow 4.08}$ | $59.18_{\uparrow 2.05}$ | $46.23_{\downarrow 2.04}$ |
| | MacNet | $76.55_{\downarrow 1.06}$ | $55.44_{\uparrow 0.95}$ | $22.94_{\downarrow 0.59}$ | $28.36_{\downarrow 0.21}$ | $60.87_{\uparrow 3.74}$ | $48.83_{\uparrow 0.56}$ |
| | G-Memory (Ours) | $88.81_{\uparrow 11.20}$ | $67.40_{\uparrow 12.91}$ | $27.77_{\uparrow 4.24}$ | $35.67_{\uparrow 7.10}$ | $66.24_{\uparrow 9.11}$ | $57.18_{\uparrow 8.91}$ |
| DyLAN 
 COLM 2024 | No-memory | $56.72_{\uparrow 0.00}$ | $55.38_{\uparrow 0.00}$ | $11.62_{\uparrow 0.00}$ | $31.69_{\uparrow 0.00}$ | $60.20_{\uparrow 0.00}$ | $43.12_{\uparrow 0.00}$ |
| | Voyager | $66.42_{\uparrow 9.70}$ | $62.83_{\uparrow 7.45}$ | $15.10_{\uparrow 3.48}$ | $32.64_{\uparrow 0.95}$ | $62.24_{\uparrow 2.04}$ | $47.85_{\uparrow 4.73}$ |
| | MemoryBank | $55.22_{\downarrow 1.50}$ | $54.74_{\downarrow 0.64}$ | $8.08_{\downarrow 3.54}$ | $29.59_{\downarrow 2.10}$ | $59.13_{\downarrow 1.07}$ | $41.35_{\downarrow 1.77}$ |
| | Generative | $67.91_{\uparrow 11.19}$ | $64.16_{\uparrow 8.78}$ | $13.87_{\uparrow 2.25}$ | $29.29_{\downarrow 2.40}$ | $62.30_{\uparrow 2.10}$ | $47.51_{\uparrow 4.39}$ |
| | MetaGPT-M | $69.40_{\uparrow 12.68}$ | $62.37_{\uparrow 6.99}$ | $14.45_{\uparrow 2.83}$ | $32.34_{\uparrow 0.65}$ | $60.20_{\uparrow 0.00}$ | $47.75_{\uparrow 4.63}$ |
| | ChatDev-M | $46.27_{\downarrow 10.45}$ | $53.35_{\downarrow 2.03}$ | $10.75_{\downarrow 0.87}$ | $22.45_{\downarrow 9.24}$ | $58.33_{\downarrow 1.87}$ | $38.23_{\downarrow 4.89}$ |
| | MacNet-M | $53.44_{\downarrow 3.28}$ | $54.32_{\downarrow 1.06}$ | $12.11_{\uparrow 0.49}$ | $30.12_{\downarrow 1.57}$ | $61.10_{\uparrow 0.90}$ | $42.22_{\downarrow 0.90}$ |
| | G-Memory (Ours) | $70.90_{\uparrow 14.18}$ | $65.64_{\uparrow 10.26}$ | $18.95_{\uparrow 7.33}$ | $34.69_{\uparrow 3.00}$ | $64.22_{\uparrow 4.02}$ | $50.88_{\uparrow 7.76}$ |
| MacNet 
 ICLR 2025 | No-memory | $51.49_{\uparrow 0.00}$ | $57.53_{\uparrow 0.00}$ | $12.18_{\uparrow 0.00}$ | $28.57_{\uparrow 0.00}$ | $60.29_{\uparrow 0.00}$ | $42.01_{\uparrow 0.00}$ |
| | Voyager | $61.94_{\uparrow 10.45}$ | $64.53_{\uparrow 7.00}$ | $14.06_{\uparrow 1.88}$ | $32.65_{\uparrow 4.08}$ | $62.54_{\uparrow 2.25}$ | $47.14_{\uparrow 5.13}$ |
| | MemoryBank | $50.00_{\downarrow 1.49}$ | $60.15_{\uparrow 2.62}$ | $8.64_{\downarrow 3.54}$ | $33.67_{\uparrow 5.10}$ | $61.22_{\uparrow 0.93}$ | $42.74_{\uparrow 0.73}$ |
| | Generative | $62.69_{\uparrow 11.20}$ | $65.49_{\uparrow 7.96}$ | $7.92_{\downarrow 4.26}$ | $29.59_{\uparrow 1.02}$ | $63.27_{\uparrow 2.98}$ | $45.79_{\uparrow 3.78}$ |
| | MetaGPT-M | $63.70_{\uparrow 12.21}$ | $65.27_{\uparrow 7.74}$ | $16.03_{\uparrow 3.85}$ | $31.00_{\uparrow 2.43}$ | $59.33_{\downarrow 0.96}$ | $47.07_{\uparrow 5.06}$ |
| | ChatDev-M | $49.25_{\downarrow 2.24}$ | $56.58_{\downarrow 0.95}$ | $13.51_{\uparrow 1.33}$ | $29.00_{\uparrow 0.43}$ | $59.18_{\downarrow 1.11}$ | $41.50_{\downarrow 0.51}$ |
| | MacNet-M | $53.44_{\uparrow 1.95}$ | $56.14_{\downarrow 1.39}$ | $13.59_{\uparrow 1.41}$ | $27.89_{\downarrow 0.68}$ | $59.20_{\downarrow 1.09}$ | $42.05_{\uparrow 0.04}$ |
| | G-Memory (Ours) | $67.16_{\uparrow 15.67}$ | $68.11_{\uparrow 10.58}$ | $24.33_{\uparrow 12.15}$ | $35.69_{\uparrow 7.12}$ | $64.44_{\uparrow 4.15}$ | $51.95_{\uparrow 9.94}$ |

## 5 Experiment

In this section, we conduct extensive experiments to answer: (**RQ1**) How does `G-Memory` perform compared to existing single/multi-agent memory architectures? (**RQ2**) Does `G-Memory` incur excessive resource overhead? (**RQ3**) How sensitive is `G-Memory` to its key components and parameters?

### 5.1 Experiment Setup

**Datasets and Benchmarks.**   To thoroughly evaluate the effectiveness of `G-Memory`, we adopt five widely-adopted benchmarks across three domains: **(1) Knowledge reasoning**, including HotpotQA [75] and FEVER [76]; **(2) Embodied action**, including ALFWorld [77] and SciWorld [78]; **(3) Game**, namely PDDL [79]. Details on these benchmarks are in Appendix A.1.

**Baselines.**   We select four representative single-agent memory baselines, including non-memory, Voyager [16], MemoryBank [36], and Generative Agents [19], as well as three multi-agent memory implementations from MetaGPT [21], ChatDev [45], and MacNet [46], denoted as MetaGPT-M, ChatDev-M, and MacNet-M, respectively. Details are in Appendix A.2.

**MAS and LLM Backbones.**   We select three representative multi-agent frameworks to integrate with `G-Memory` and the baselines, including **AutoGen** [13], **DyLAN** [71], and **MacNet** [46]. More details on the MAS setups are placed in Appendix A.3. For instantiating these MAS frameworks, we adopt two open-source LLMs, `Qwen-2.5-7b` and `Qwen-2.5-14b`, as well as one proprietary LLM, `gpt-4o-mini`. The deployment of `Qwen` series is via local instantiation using Ollama[1], and `GPT` models are accessed via OpenAI APIs.

**Parameter Configurations.**   We implement the embedding function $\mathbf{v}(\cdot)$ in Equation (4) with `ALL-MINILM-L6-v2` [80]. The number of the most relevant interaction graphs $M$ in Equation (7) is set among $\{2, 3, 4, 5\}$, and the number of relevant queries $k$ in Equation (4) is set among $\{1, 2\}$. The detailed ablation study on hyper-parameters is placed in Section 5.4.

### 5.2 Main Results (RQ1)

Tables 1, 2 and 3 comprehensively report the performance of different memory architectures across three LLM backbones and three MAS frameworks. We summarize the key observations as follows:

---

[1] http://github.com/ollama/ollama

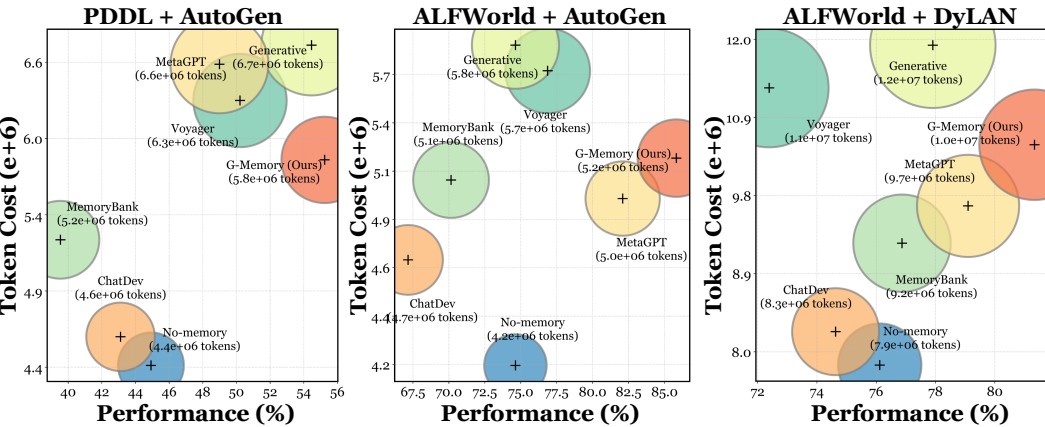

Figure 3: Cost analysis of `G-Memory`. We showcase the performance versus the overall system token cost when combined with different memory architectures.

**Takeaway ❶: `G-Memory` consistently improves performance across all task domains and MAS frameworks.** As shown in Table 2, when integrated with AutoGen and MacNet (powered by `Qwen-2.5-7b`), `G-Memory` surpasses the best-performing single-/multi-agent memory baselines by an average of 6.8% and 5.5%, respectively. With the more capable `Qwen-2.5-14b`, the improvement is even more pronounced: in Table 3, `G-Memory` boosts MacNet's performance on ALFWorld from 58.21% to 79.10%, achieving a substantial 20.89% gain.

**Takeaway ❷: Multi-agent systems demand specialized memory designs.** A thorough examination of existing baselines reveals a surprising insight: most memory mechanisms fail to consistently benefit MAS settings. In Table 2, baselines such as Voyager and MemoryBank degrade AutoGen's performance on PDDL by as much as 4.17% and 1.34%, respectively. We attribute this to the inability of these methods to provide agent role-specific memory support, which is essential in the PDDL strategic game tasks, where effective division of labor is critical to success. Even MAS-oriented designs, such as ChatDev-M, result in a 2.32% performance drop when applied to MacNet+SciWorld. We attribute this to ChatDev-M's narrow memory scope—storing only the execution results of past queries, which provides limited utility in embodied action environments. These findings highlight the necessity of `G-Memory`'s core characteristics: role-specific memory cues, abstracted high-level insights, and trajectory condensation—all of which are critical for effective memory in MAS.

### 5.3 Cost Analysis (RQ2)

To evaluate the efficiency of `G-Memory` in terms of token consumption, we visualize the performance versus token cost trade-off across various settings, as shown in Figures 3 and 7. Our findings are:

**Takeaway ❸: `G-Memory` achieves high-performing collective memory without excessive token consumption.** As depicted in Figure 3, `G-Memory` consistently delivers the highest performance improvement (10.32% ↑ over no-memory setting on PDDL+AutoGen) while maintaining a modest increase in token consumption (only $1.4 \times 10^6$). In contrast, MetaGPT-M incurred an additional $2.2 \times 10^6$ tokens for a mere 4.07% gain. This clearly demonstrates the token-efficiency of `G-Memory`.

### 5.4 Framework Analysis (RQ3)

**Sensitivity Analysis.** Regarding the hop expansion, as shown in Figure 4a, 1-hop expansion consistently yields the best or near-best performance across tasks, with peak accuracies of 85.82% (ALFWorld), 55.24% (PDDL) in AutoGen. In contrast, 2-hop and 3-hop settings often degrade performance, *e.g.*, PDDL drops to 49.79% (2-hop). This suggests that excessive hop expansion may introduce irrelevant insights during memory upward traversal, impairing task-specific reasoning. Similarly, Figure 4b shows that the optimal $k$ is among $\{1, 2\}$. Larger $k$ values (*e.g.*, $k=5$) can significantly degrade the system performance, *e.g.*, 7.71% ↓ on ALFWorld+AutoGen and 2.5% ↓ on FEVER+DyLAN, indicating that retrieving more queries may introduce task-irrelevant noise. Collectively, we employ 1-hop expansion and $k \in \{1, 2\}$ throughout the experiments.

**Ablation Study.** Figure 4c presents an ablation of `G-Memory` by isolating the impact of the high-level insight module ($\mathcal{I}^{\mathcal{S}}$ in Equation (6)) and fine-grained interactions ($\{\hat{\mathcal{G}}_{\text{inter}}^{Q_i}\}_{i=1}^{|M|}$ in Equation (7)). As shown, removing either part leads to a consistent performance drop. When only fine-grained interactions are enabled, the average scores drop by 4.47% ↓ for AutoGen and 3.82% ↓ for DyLAN

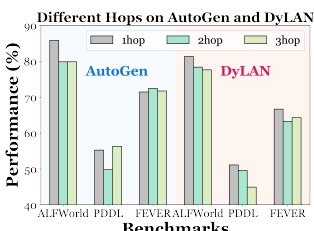

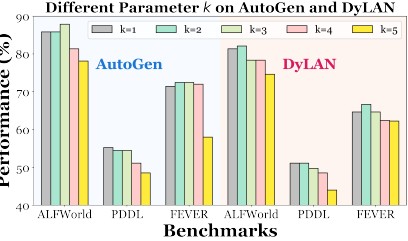

| MAS | Inter. | Insi. | PDDL | FEVER |
|---|:---:|:---:|---|---|
| AutoGen | ✔ | ○ | 54.46 | 63.27 |
|  | ○ | ✔ | 50.00 | 68.77 |
|  | ✔ | ✔ | 55.24 | 71.43 |
| DyLAN | ✔ | ○ | 48.75 | 61.39 |
|  | ○ | ✔ | 46.69 | 64.31 |
|  | ✔ | ✔ | 51.12 | 66.66 |

(a) Sensitivity analysis on #hop.   (b) Sensitivity analysis on parameter $k$.   (c) Ablation study on two variants of `G-Memory`.

Figure 4: (a) Sensitivity analysis of the hop expansion in Equation (5); (b) Sensitivity analysis of the number of selected queries $k$ in Equation (4); (c) We study two variants of `G-Memory`: merely providing high-level insights (*i.e.*, the insights $\mathcal{I}^{\mathcal{S}}$ in Equation (6)) or fine-grained interactions (*i.e.*, the core trajectories in Equation (7)). All the experiments here are done with `Qwen-2.5-14b`.

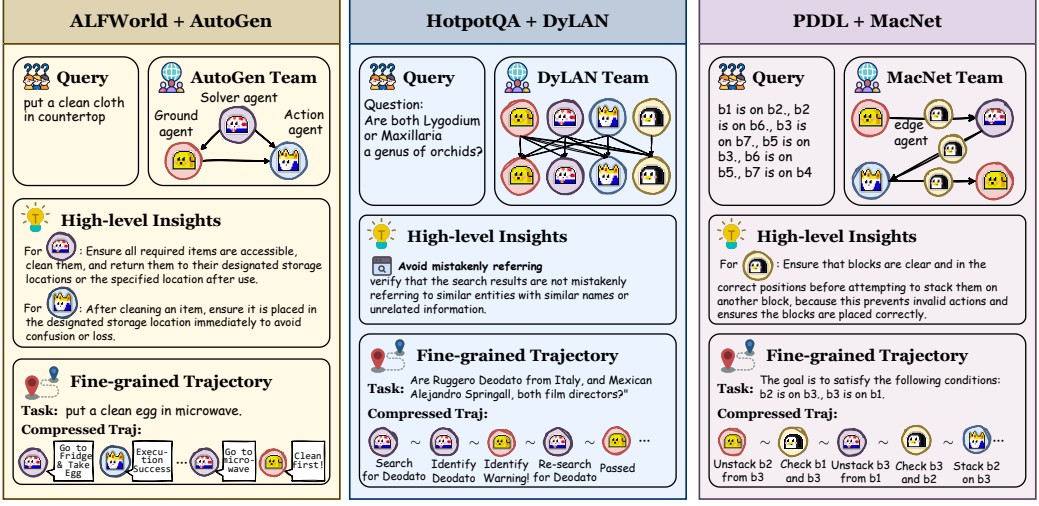

Figure 5: Case study of `G-Memory`.

compared to the full method. Conversely, enabling only insights leads to smaller drops of $3.95\%$ and $3.39\%$. This indicates that while both components are contributive, interactions offer a slightly greater impact, likely due to their preserving more fine-grained, dialogue-level contextual grounding.

## 5.5 Case Study

Figure 5 illustrates concrete memory cues provided by `G-Memory` across diverse tasks. For example, in the ALFWorld+AutoGen setting, given the task query "put a clean cloth in countertop", `G-Memory` successfully retrieves a highly analogous historical query, "put a clean egg in microwave"—both requiring the object to be in a clean state. Alongside this, `G-Memory` surfaces a critical trajectory segment where the solver agent attempts to place the egg in the microwave before cleaning, prompting the ground agent to intervene. This collaborative trajectory offers actionable guidance for the current task. Moreover, the high-level insights retrieved by `G-Memory` prove equally valuable for task execution. In the context of HotpotQA's web search task, `G-Memory` retrieves an insight warning against "mistakenly referring", which helps prevent agents from incorrectly answering based on similarly named individuals. Overall, `G-Memory` provides effective multi-level memory support across varied domains, including embodied action, knowledge reasoning, and game environments.

## 6 Conclusion & Limitation

In this paper, we conduct a thorough examination of existing memory architectures designed for multi-agent systems (MAS) and identify that their overly simplified designs fundamentally hinder the systems' capacity for self-evolution. To bridge this gap, we propose `G-Memory`, a hierarchical memory framework that organizes the complex and extended interaction trajectories of MAS into a three-tier graph hierarchy: the *insight*, *query*, and *interaction* graphs. `G-Memory` provides each agent with customized and hierarchical memory cues, ranging from abstract, generalizable insights

to fine-grained, task-critical collaborative segments, and dynamically evolves its knowledge base across episodes. Extensive experiments demonstrate that `G-Memory` can be seamlessly integrated into state-of-the-art MAS frameworks, significantly enhancing their self-evolution capability, *e.g.*, up to $20.89\% \uparrow$ improvement on embodied action tasks. **Limitations:** Although `G-Memory` has been evaluated across three domains and five benchmarks, further validation on more diverse tasks (*e.g.*, medical QA) would strengthen its soundness, which we leave for future work.

## Acknowledgment

This research is supported in part by NUS Start-up Grant A-0010106-00-00.

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

## Impact Statement

`G-Memory` introduces a structured, hierarchical memory architecture for multi-agent systems (MAS), enabling large language model (LLM)-based agents to store, recall, and reason over past experiences with enhanced task generalization and cooperation efficiency. The broader impacts of this work include advancing the development of scalable and adaptive collective intelligence, with potential applications in long-term robotic planning, real-world decision-making systems, and collaborative AI assistants. However, if the underlying language model is compromised or adversarially manipulated, the memory mechanisms could amplify incorrect reasoning. We urge responsible deployment of this architecture with appropriate safeguards, including continual validation, adversarial robustness checks, and alignment with human values.

## A    Experimental Details

### A.1    Dataset Descriptions

In this section, we describe the datasets used in our experiments:

- **ALFWorld** [77] (available at https://alfworld.github.io/, MIT license) is a text-based embodied environment featuring household tasks, where agents navigate and interact with objects via natural language commands.
- **ScienceWorld** [78] (available at https://github.com/allenai/ScienceWorld, Apache-2.0 license) is another text-based embodied environment designed for interactive science tasks. Agents must navigate rooms and conduct experiments, testing their ability to perform procedural reasoning and scientific exploration.
- **PDDL** is a game dataset from AgentBoard [79] (available at https://github.com/hkust-nlp/AgentBoard, Custom properties), comprising a variety of strategic games where agents use PDDL expressions to complete complex tasks.
- **HotpotQA** [75] (available at https://hotpotqa.github.io/, CC BY-SA 4.0 License) is a multi-hop question answering dataset with strong supervision on supporting facts. It evaluates the agent's ability to retrieve and synthesize information, especially through web search tools, for explainable reasoning.
- **FEVER** [76] (available at https://fever.ai/dataset/fever.html, Creative Commons Attribution-ShareAlike License) is a knowledge-intensive dataset focused on fact verification. Agents must validate claims using web search APIs, making it a benchmark for evidence-based reasoning.

**Evaluation Metrics.**    We use *exact match* accuracy for FEVER and HotpotQA. For ScienceWorld and PDDL, we report the *progress rate*, and for ALFWorld, we use the *success rate* as the evaluation metric.

## A.2 Baseline Setup

In this section, we provide detailed descriptions of each baseline used in our comparison:

- **Voyager**: The Voyager memory is derived from the Voyager agent [16], where an embodied agent continuously interacts with the Minecraft environment and creates new artifacts. Memory serves as the core driver of the agent's evolution. As Voyager's memory design is tailored for a single-agent setting, we adapt it to the multi-agent scenario by implementing agent-specific history retrieval based on each agent's visible dialogue context. Other single-agent memory designs are adapted in a similar manner.

- **MemoryBank**: MemoryBank [36] mimics anthropomorphic memory behaviors by selectively preserving and forgetting information. It incorporates a memory updating mechanism inspired by the Ebbinghaus Forgetting Curve, allowing the agent to reinforce or discard memory based on temporal decay and the relative importance of stored information.

- **Generative**: This memory baseline is based on [19], which includes both raw observational memory and high-level reflective memory. The latter captures abstract thoughts generated by the agent through reflection, providing a more structured and conceptualized representation of experience.

- **MetaGPT-M**: The memory design originates from MetaGPT [21], focusing solely on *inside-trial* memory—information stored internally during the resolution of a single task by multiple agents.

- **ChatDev-M**: This memory design is adapted from ChatDev [45], which incorporates both *inside-trial* and *cross-trial* memory. The inside-trial memory is passed from the central or initiating agent at the beginning of each round to provide guidance based on prior interactions. The cross-trial memory is relatively simple, storing past solutions to previous queries for future retrieval. However, in our task, it does not effectively manage the information-rich inter-agent collaboration.

- **MacNet-M**: This memory design is adopted from MacNet [46], where the *inside-trial* memory consists solely of the final answers generated in the previous round. All non-artifact dialogue contexts, *i.e.*, the interaction trajectories among agents, are entirely discarded.

## A.3 Multi-agent System Setup

In this section, we detail the setups of our three adopted MAS frameworks, AutoGen, DyLAN and MacNet:

### A.3.1 AutoGen

AutoGen [13] is a popular multi-agent orchestration framework, to coordinate interactions among specialized agents for problem-solving tasks. Specifically, we utilize their A3 : Decision Making structure, which is composed of: (1) a **Solver Agent**, responsible for generating solutions, initialized with the system prompt "You are a smart agent designed to solve problems."; (2) a **Ground Truth Agent**, which critically evaluates the solver's output and identifies potential errors based on a reference standard; and (3) an **Executor Agent**, tasked with translating validated solutions into executable commands. This modular design enables transparent, verifiable, and actionable multi-agent collaboration.

### A.3.2 DyLAN

DyLAN [71] is a debate-style framework similar to LLM-Debate, but incorporates a more efficient agent-wise early stopping mechanism during multi-turn interactions. DyLAN utilizes an agent selection algorithm based on an unsupervised metric, namely the *Agent Importance Score*, which identifies the most contributive agents through a preliminary trial tailored to the specific task. In our implementation of DyLAN, three agents engage in the debate, while an additional ranker agent evaluates their relative importance.

### A.4 MacNet

MacNet [46] is a representative work that explores decentralized and scalable multi-agent systems. Its key feature lies in the absence of a central agent; instead, it introduces *edge agents*, which are invoked between agent interactions to provide actionable instructions to the next agent based on the previous agent's outputs. In our implementation, we adopt the random graph topology from MacNet, shown to be robust across diverse scenarios, and employ five agents in addition to the edge agents.

## B  Additional Experiment Results

### B.1  RQ1 Results

Tables 2 and 3 present additional experimental results using `Qwen-2.5-7b` and `Qwen-2.5-14b` as the LLM backbones. Appendix B.1 illustrates the success rate curves on ALFWorld as the number of trials increases, comparing different MAS frameworks combined with various memory architectures. As shown in Figures 6b and 6c, `G-Memory` consistently enables MAS frameworks to achieve success with fewer trials and leads to higher final performance ceilings.

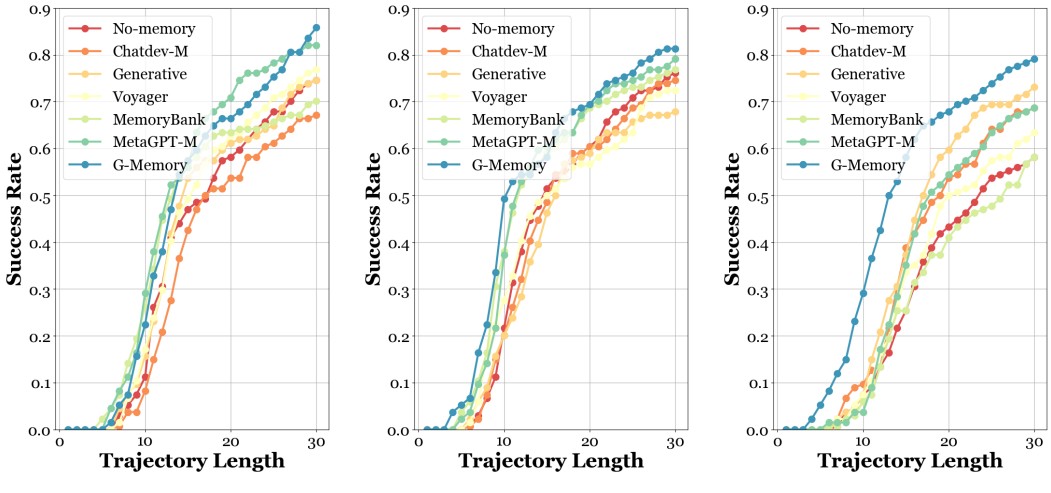

(a) The performance trajectory of AutoGen on ALFWorld.

(b) The performance trajectory of DyLAN on ALFWorld.

(c) The performance trajectory of MacNet on ALFWorld.

### B.2  RQ2 Results

Figure 7 provides additional comparisons of token cost across various benchmarks and MAS frameworks when combined with different memory architectures. Overall, `G-Memory` incurs only a marginal or no increase in token cost compared to classical baselines such as Generative and MetaGPT-M, while consistently delivering the most significant performance improvements.

### B.3  Latency Results

As shown in Table 4, `G-Memory` incurs only moderate inference overhead even when delivering the most significant performance gains. For instance, on AutoGen+ALFWorld, it increases latency by merely $9\%$ compared to the no-memory baseline; (2) in some cases, `G-Memory` even improves efficiency, *e.g.*, on DyLAN+ALFWorld, it leads to a $20\%$ speed-up, as helpful memory cues enable the multi-agent system to reach correct actions more quickly and terminate the interaction earlier.

Table 2: Performance comparison with single/multi-agent memory architectures on five benchmarks. The underlying LLM backbone is `Qwen-2.5-7b`. We highlight the best and second best results.

| MAS | Memory | ALFWorld | SciWorld | PDDL | HotpotQA | FEVER | Avg. |
|---|---|---|---|---|---|---|---|
| Vanilla LLM | No-memory | $37.31_{\uparrow 0.00}$ | $23.49_{\uparrow 0.00}$ | $10.86_{\uparrow 0.00}$ | $20.26_{\uparrow 0.00}$ | $48.17_{\uparrow 0.00}$ | $28.02_{\uparrow 0.00}$ |
| | Voyager | $38.19_{\uparrow 0.88}$ | $24.11_{\uparrow 0.62}$ | $12.14_{\uparrow 1.28}$ | $19.12_{\downarrow 1.14}$ | $49.68_{\uparrow 1.51}$ | $28.65_{\uparrow 0.63}$ |
| | MemoryBank | $40.30_{\uparrow 2.99}$ | $21.64_{\downarrow 1.85}$ | $14.36_{\uparrow 3.50}$ | $18.79_{\downarrow 1.47}$ | $47.66_{\downarrow 0.51}$ | $28.55_{\uparrow 0.53}$ |
| | Generative | $39.16_{\uparrow 1.85}$ | $26.10_{\uparrow 2.61}$ | $11.37_{\uparrow 0.51}$ | $23.48_{\uparrow 3.22}$ | $52.50_{\uparrow 4.33}$ | $30.52_{\uparrow 2.50}$ |
| AutoGen COLM 2024 | No-memory | $52.99_{\uparrow 0.00}$ | $30.27_{\uparrow 0.00}$ | $16.17_{\uparrow 0.00}$ | $33.33_{\uparrow 0.00}$ | $58.74_{\uparrow 0.00}$ | $38.30_{\uparrow 0.00}$ |
| | Voyager | $55.22_{\uparrow 2.23}$ | $26.70_{\downarrow 3.57}$ | $12.00_{\downarrow 4.17}$ | $34.29_{\uparrow 0.96}$ | $52.44_{\downarrow 6.30}$ | $36.13_{\downarrow 2.17}$ |
| | MemoryBank | $53.37_{\uparrow 0.38}$ | $27.33_{\downarrow 2.94}$ | $14.83_{\downarrow 1.34}$ | $32.67_{\downarrow 0.66}$ | $59.45_{\uparrow 0.71}$ | $37.53_{\downarrow 0.77}$ |
| | Generative | $62.69_{\uparrow 9.70}$ | $31.45_{\uparrow 1.18}$ | $17.88_{\uparrow 1.71}$ | $34.17_{\uparrow 0.84}$ | $61.25_{\uparrow 2.51}$ | $41.49_{\uparrow 3.19}$ |
| | MetaGPT-M | $55.52_{\uparrow 2.53}$ | $32.44_{\uparrow 2.17}$ | $17.04_{\uparrow 0.87}$ | $35.36_{\uparrow 2.03}$ | $63.33_{\uparrow 4.59}$ | $40.74_{\uparrow 2.44}$ |
| | ChatDev-M | $46.27_{\downarrow 6.72}$ | $28.67_{\downarrow 1.60}$ | $13.42_{\downarrow 2.75}$ | $31.11_{\downarrow 2.22}$ | $61.32_{\uparrow 2.58}$ | $36.16_{\downarrow 2.14}$ |
| | MacNet-M | $53.18_{\uparrow 0.19}$ | $31.10_{\uparrow 0.83}$ | $16.89_{\uparrow 0.72}$ | $34.29_{\uparrow 0.96}$ | $58.43_{\downarrow 0.31}$ | $38.78_{\uparrow 0.48}$ |
| | G-Memory (Ours) | $67.91_{\uparrow 14.92}$ | $34.89_{\uparrow 4.62}$ | $21.01_{\uparrow 4.84}$ | $37.34_{\uparrow 4.01}$ | $64.34_{\uparrow 5.60}$ | $45.10_{\uparrow 6.80}$ |
| DyLAN COLM 2024 | No-memory | $41.34_{\uparrow 0.00}$ | $29.84_{\uparrow 0.00}$ | $13.56_{\uparrow 0.00}$ | $24.29_{\uparrow 0.00}$ | $56.23_{\uparrow 0.00}$ | $33.05_{\uparrow 0.00}$ |
| | Voyager | $51.49_{\uparrow 10.15}$ | $26.66_{\downarrow 3.18}$ | $10.62_{\downarrow 2.94}$ | $26.23_{\uparrow 1.94}$ | $55.39_{\downarrow 0.84}$ | $34.08_{\uparrow 1.03}$ |
| | MemoryBank | $46.46_{\uparrow 5.12}$ | $26.99_{\downarrow 2.85}$ | $14.10_{\uparrow 0.54}$ | $22.44_{\downarrow 1.85}$ | $59.21_{\uparrow 2.98}$ | $33.84_{\uparrow 0.79}$ |
| | Generative | $48.52_{\uparrow 7.18}$ | $31.55_{\uparrow 1.71}$ | $16.31_{\uparrow 2.75}$ | $26.54_{\uparrow 2.25}$ | $50.19_{\downarrow 6.04}$ | $34.62_{\uparrow 1.57}$ |
| | MetaGPT-M | $42.54_{\uparrow 1.20}$ | $30.93_{\uparrow 1.09}$ | $14.47_{\uparrow 0.91}$ | $19.33_{\downarrow 4.96}$ | $57.22_{\uparrow 0.99}$ | $32.90_{\downarrow 0.15}$ |
| | ChatDev-M | $39.85_{\downarrow 1.49}$ | $28.25_{\downarrow 1.59}$ | $7.14_{\downarrow 6.42}$ | $17.32_{\downarrow 6.97}$ | $50.67_{\downarrow 5.56}$ | $28.65_{\downarrow 4.41}$ |
| | MacNet-M | $42.48_{\uparrow 1.14}$ | $28.22_{\downarrow 1.62}$ | $14.23_{\uparrow 0.67}$ | $25.12_{\uparrow 0.83}$ | $55.34_{\downarrow 0.89}$ | $33.08_{\uparrow 0.03}$ |
| | G-Memory (Ours) | $52.99_{\uparrow 11.65}$ | $33.81_{\uparrow 3.97}$ | $20.71_{\uparrow 7.15}$ | $29.33_{\uparrow 5.04}$ | $63.67_{\uparrow 7.44}$ | $40.10_{\uparrow 7.05}$ |
| MacNet ICLR 2025 | No-memory | $44.03_{\uparrow 0.00}$ | $28.76_{\uparrow 0.00}$ | $13.36_{\uparrow 0.00}$ | $22.24_{\uparrow 0.00}$ | $55.12_{\uparrow 0.00}$ | $32.70_{\uparrow 0.00}$ |
| | Voyager | $47.01_{\uparrow 2.98}$ | $28.88_{\uparrow 0.12}$ | $11.36_{\downarrow 2.00}$ | $25.67_{\uparrow 3.43}$ | $58.78_{\uparrow 3.66}$ | $34.34_{\uparrow 1.64}$ |
| | MemoryBank | $52.24_{\uparrow 8.21}$ | $27.86_{\downarrow 0.90}$ | $13.33_{\downarrow 0.03}$ | $23.97_{\uparrow 1.73}$ | $54.18_{\downarrow 0.94}$ | $34.32_{\uparrow 1.61}$ |
| | Generative | $48.51_{\uparrow 4.48}$ | $31.05_{\uparrow 2.29}$ | $14.04_{\uparrow 0.68}$ | $24.49_{\uparrow 2.25}$ | $56.08_{\uparrow 0.96}$ | $34.83_{\uparrow 2.13}$ |
| | MetaGPT-M | $52.99_{\uparrow 8.96}$ | $29.87_{\uparrow 1.11}$ | $16.58_{\uparrow 3.22}$ | $25.51_{\uparrow 3.27}$ | $53.88_{\downarrow 1.24}$ | $35.77_{\uparrow 3.06}$ |
| | ChatDev-M | $44.78_{\uparrow 0.75}$ | $26.44_{\downarrow 2.32}$ | $10.19_{\downarrow 3.17}$ | $16.32_{\downarrow 5.92}$ | $56.02_{\uparrow 0.90}$ | $30.75_{\downarrow 1.95}$ |
| | MacNet-M | $43.55_{\downarrow 0.48}$ | $30.11_{\uparrow 1.35}$ | $12.91_{\downarrow 0.45}$ | $21.77_{\downarrow 0.47}$ | $50.71_{\downarrow 4.41}$ | $31.81_{\downarrow 0.89}$ |
| | G-Memory (Ours) | $54.48_{\uparrow 10.45}$ | $32.23_{\uparrow 3.47}$ | $17.48_{\uparrow 4.12}$ | $27.53_{\uparrow 5.29}$ | $59.14_{\uparrow 4.02}$ | $38.17_{\uparrow 5.47}$ |

## B.4 Case Study

### B.4.1 Case Study on Insight Graphs

Figure 8 visualizes the high-level insights summarized by `G-Memory` on the ALFWorld benchmark across different MAS frameworks and LLM backbones. Given that ALFWorld naturally consists of diverse task categories, we further examine how insight nodes corresponding to different task types are interconnected. Overall, we observe dense intra-category connections among insights derived from similar tasks, while also noting the emergence of meaningful inter-category links, reflecting transferable patterns across task domains.

### B.4.2 Case Study on Query Graphs

Figures 9 to 11 visualize the query graphs constructed by `G-Memory` on the ALFWorld, PDDL, and SciWorld benchmarks. Recall that a directed edge between two query nodes indicates that the historical trajectory of one query offers useful guidance for the execution of another. We observe emergent clustering patterns, where groups of semantically similar queries form densely connected subgraphs, while sparser inter-cluster edges capture cross-task inspirations. These patterns demonstrate `G-Memory`'s ability to effectively organize and relate collaborative experiences through structured memory reasoning.

Table 3: Performance comparison with single/multi-agent memory architectures on five benchmarks. The underlying LLM backbone is `Qwen-2.5-14b`. We highlight the best and second best results.

| MAS | Memory | ALFWorld | SciWorld | PDDL | HotpotQA | FEVER | Avg. |
|---|---|---|---|---|---|---|---|
| AutoGen COLM 2024 | No-memory | 74.63$_{\uparrow 0.00}$ | 46.84$_{\uparrow 0.00}$ | 44.92$_{\uparrow 0.00}$ | 24.49$_{\uparrow 0.00}$ | 63.27$_{\uparrow 0.00}$ | 50.83$_{\uparrow 0.00}$ |
| | Voyager | 76.87$_{\uparrow 2.24}$ | 59.00$_{\uparrow 12.16}$ | 50.21$_{\uparrow 5.29}$ | 31.33$_{\uparrow 6.84}$ | 61.22$_{\downarrow 2.05}$ | 55.73$_{\uparrow 4.90}$ |
| | MemoryBank | 70.15$_{\downarrow 4.48}$ | 54.18$_{\uparrow 7.34}$ | 39.54$_{\downarrow 5.38}$ | 32.65$_{\uparrow 8.16}$ | 64.29$_{\uparrow 1.02}$ | 52.16$_{\uparrow 1.33}$ |
| | Generative | 74.63$_{\uparrow 0.00}$ | 57.37$_{\uparrow 10.53}$ | 54.46$_{\uparrow 9.54}$ | 33.21$_{\uparrow 8.72}$ | 63.27$_{\uparrow 0.00}$ | 56.59$_{\uparrow 5.76}$ |
| | MetaGPT-M | 82.09$_{\uparrow 7.46}$ | 58.86$_{\uparrow 12.02}$ | 48.99$_{\uparrow 4.07}$ | 31.63$_{\uparrow 7.14}$ | 62.27$_{\downarrow 1.00}$ | 56.77$_{\uparrow 5.94}$ |
| | ChatDev-M | 67.16$_{\downarrow 7.47}$ | 40.69$_{\downarrow 6.15}$ | 43.11$_{\downarrow 1.81}$ | 31.77$_{\uparrow 7.28}$ | 61.28$_{\downarrow 1.99}$ | 48.80$_{\downarrow 2.03}$ |
| | MacNet-M | 73.65$_{\downarrow 0.98}$ | 42.14$_{\downarrow 4.70}$ | 45.94$_{\uparrow 1.02}$ | 26.72$_{\uparrow 2.23}$ | 64.69$_{\uparrow 1.42}$ | 50.63$_{\downarrow 0.20}$ |
| | G-Memory (Ours) | 85.82$_{\uparrow 11.19}$ | 60.62$_{\uparrow 13.78}$ | 55.24$_{\uparrow 10.32}$ | 34.61$_{\uparrow 10.12}$ | 71.43$_{\uparrow 8.16}$ | 61.54$_{\uparrow 10.71}$ |
| DyLAN COLM 2024 | No-memory | 76.12$_{\uparrow 0.00}$ | 53.24$_{\uparrow 0.00}$ | 41.83$_{\uparrow 0.00}$ | 30.61$_{\uparrow 0.00}$ | 63.34$_{\uparrow 0.00}$ | 53.03$_{\uparrow 0.00}$ |
| | Voyager | 72.39$_{\downarrow 3.73}$ | 58.93$_{\uparrow 5.69}$ | 48.54$_{\uparrow 6.71}$ | 30.71$_{\uparrow 0.10}$ | 65.31$_{\uparrow 1.97}$ | 55.18$_{\uparrow 2.15}$ |
| | MemoryBank | 76.87$_{\uparrow 0.75}$ | 57.92$_{\uparrow 4.68}$ | 39.65$_{\downarrow 2.18}$ | 29.59$_{\downarrow 1.02}$ | 63.25$_{\downarrow 0.09}$ | 53.46$_{\uparrow 0.43}$ |
| | Generative | 77.91$_{\uparrow 1.79}$ | 61.52$_{\uparrow 8.28}$ | 46.69$_{\uparrow 4.86}$ | 31.33$_{\uparrow 0.72}$ | 61.39$_{\downarrow 1.95}$ | 55.77$_{\uparrow 2.74}$ |
| | MetaGPT-M | 79.10$_{\uparrow 2.98}$ | 61.29$_{\uparrow 8.05}$ | 49.75$_{\uparrow 7.92}$ | 28.61$_{\downarrow 2.00}$ | 64.11$_{\uparrow 0.77}$ | 56.57$_{\uparrow 3.54}$ |
| | ChatDev-M | 74.63$_{\downarrow 1.49}$ | 54.03$_{\uparrow 0.79}$ | 44.44$_{\uparrow 2.61}$ | 30.67$_{\uparrow 0.06}$ | 62.25$_{\downarrow 1.09}$ | 53.20$_{\uparrow 0.18}$ |
| | MacNet-M | 72.77$_{\downarrow 3.35}$ | 52.22$_{\downarrow 1.02}$ | 42.98$_{\uparrow 1.15}$ | 29.22$_{\downarrow 1.39}$ | 62.69$_{\downarrow 0.65}$ | 51.98$_{\downarrow 1.05}$ |
| | G-Memory (Ours) | 81.34$_{\uparrow 5.22}$ | 64.68$_{\uparrow 11.44}$ | 51.12$_{\uparrow 9.29}$ | 34.63$_{\uparrow 4.02}$ | 66.66$_{\uparrow 3.32}$ | 59.69$_{\uparrow 6.66}$ |
| MacNet ICLR 2025 | No-memory | 58.21$_{\uparrow 0.00}$ | 52.21$_{\uparrow 0.00}$ | 41.74$_{\uparrow 0.00}$ | 28.60$_{\uparrow 0.00}$ | 64.65$_{\uparrow 0.00}$ | 49.08$_{\uparrow 0.00}$ |
| | Voyager | 63.43$_{\uparrow 5.22}$ | 60.24$_{\uparrow 8.03}$ | 43.95$_{\uparrow 2.21}$ | 29.67$_{\uparrow 1.07}$ | 62.24$_{\downarrow 2.41}$ | 51.91$_{\uparrow 2.82}$ |
| | MemoryBank | 62.21$_{\uparrow 4.00}$ | 55.52$_{\uparrow 3.31}$ | 38.26$_{\downarrow 3.48}$ | 26.53$_{\downarrow 2.07}$ | 65.22$_{\uparrow 0.57}$ | 49.55$_{\uparrow 0.47}$ |
| | Generative | 73.13$_{\uparrow 14.92}$ | 60.83$_{\uparrow 8.62}$ | 44.00$_{\uparrow 2.26}$ | 30.53$_{\uparrow 1.93}$ | 65.31$_{\uparrow 0.66}$ | 54.76$_{\uparrow 5.68}$ |
| | MetaGPT-M | 70.43$_{\uparrow 12.22}$ | 59.70$_{\uparrow 7.49}$ | 42.34$_{\uparrow 0.60}$ | 26.26$_{\downarrow 2.34}$ | 66.33$_{\uparrow 1.68}$ | 53.01$_{\uparrow 3.93}$ |
| | ChatDev-M | 68.66$_{\uparrow 10.45}$ | 45.98$_{\downarrow 6.23}$ | 42.19$_{\uparrow 0.45}$ | 29.49$_{\downarrow 0.89}$ | 59.18$_{\downarrow 5.47}$ | 49.10$_{\uparrow 0.02}$ |
| | MacNet-M | 60.45$_{\uparrow 2.24}$ | 51.14$_{\downarrow 1.07}$ | 39.22$_{\downarrow 2.52}$ | 28.77$_{\uparrow 0.17}$ | 62.42$_{\downarrow 2.23}$ | 48.40$_{\downarrow 0.68}$ |
| | G-Memory (Ours) | 79.10$_{\uparrow 20.89}$ | 61.74$_{\uparrow 9.53}$ | 45.76$_{\uparrow 4.02}$ | 32.33$_{\uparrow 3.73}$ | 70.33$_{\uparrow 5.68}$ | 57.85$_{\uparrow 8.77}$ |

Table 4: Performance (%) and latency (s) comparison of different memory mechanisms on AutoGen and DyLAN frameworks, along with ALFWorld and SciWorld benchmarks.

| Method | AutoGen | | | | DyLAN | | | |
|---|---|---|---|---|---|---|---|---|
| | ALFWorld | | SciWorld | | ALFWorld | | SciWorld | |
| | Perf. | Lat. | Perf. | Lat. | Perf. | Lat. | Perf. | Lat. |
| No-memory | 77.61 | 19204 | 54.59 | 16953 | 56.72 | 33520 | 55.38 | 32408 |
| Voyager | 85.07 | 21754 | 62.36 | 16650 | 66.42 | 29628 | 62.83 | 31633 |
| MemoryBank | 74.96 | 15492 | 53.11 | 10104 | 55.22 | 31813 | 54.74 | 32925 |
| Generative | 86.36 | 20682 | 61.19 | 16674 | 67.91 | 31010 | 64.16 | 34038 |
| MetaGPT-M | 81.34 | 16021 | 61.91 | 15853 | 69.40 | 22936 | 62.37 | 32049 |
| ChatDev-M | 79.85 | 22347 | 50.96 | 12904 | 46.27 | 29739 | 53.35 | 33111 |
| MacNet-M | 76.55 | 21089 | 55.44 | 16882 | 53.44 | 24991 | 54.32 | 34815 |
| **G-Memory (Ours)** | **88.81** | 21113 | **67.40** | 17326 | **70.90** | 26726 | **65.64** | 33447 |

# C  Prompt Set

**Query Relevance Filtration**

```
task_relevency_system_prompt = """You are an agent designed to score the relevance
    between two pieces of text."""
task_relevency_user_prompt = """You will be given a successful case where you
    successfully complete the task. Then you will be given an ongoing task. Do
    not summarize these two cases, but rather evaluate how relevant and helpful
    the successful case is for the ongoing task, on a scale of 1-10.
Success Case:
{trajectory}
Ongoing task:
{query_scenario}
```

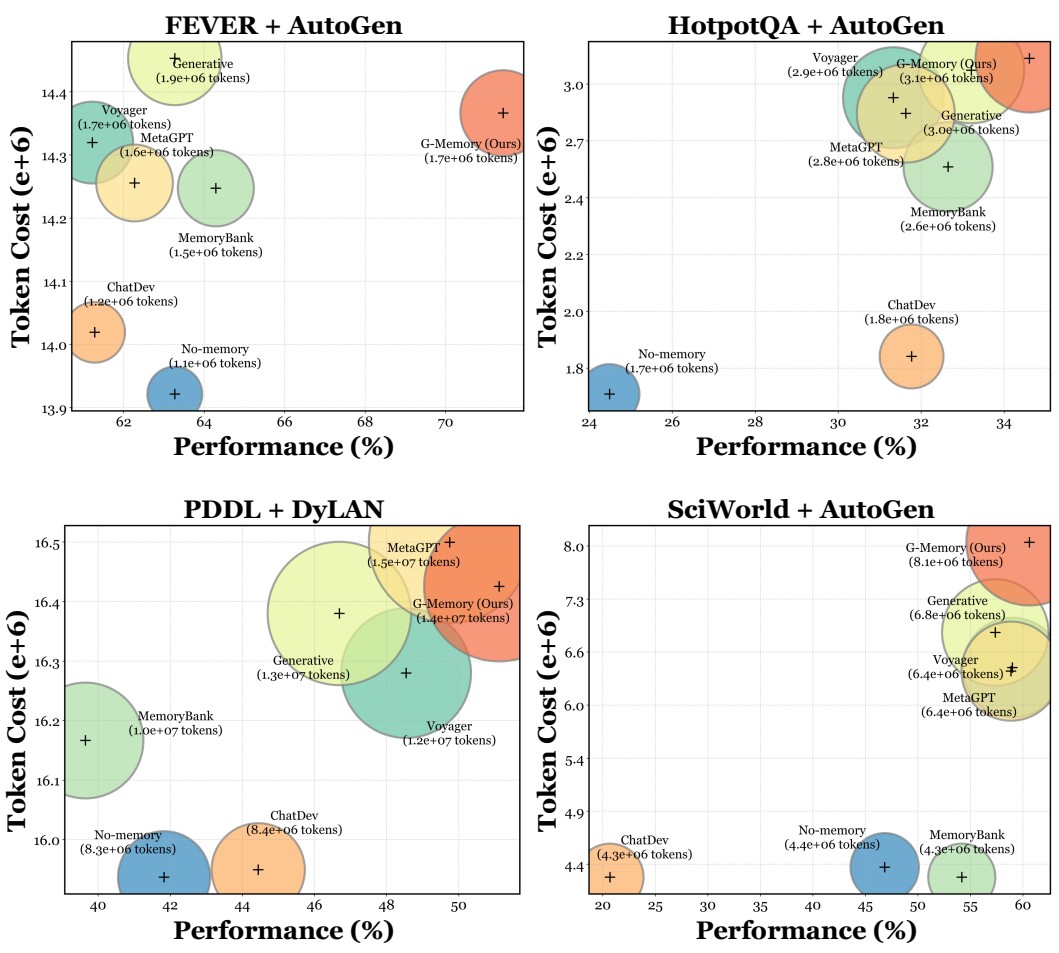

Figure 7: Cost analysis of `G-Memory`. We showcase the performance versus the overall system token cost when combined with different memory architectures.

```
Score: """
```

Graph Sparsifier

```
extract_true_traj_system_prompt = """You are an agent skilled at extracting key
    points.
Given a task and a successful execution trajectory, your job is to identify the
    critical steps needed to complete the task while filtering out less important
     steps."""

extract_true_traj_user_prompt = """
Note:
- Strictly follow the original trajectory; absolutely no steps that are not in the
     trajectory should be added.
- Even in a successful trajectory, there may be some incorrect steps. Pay
    attention to actions that correspond to "Nothing happens" observations, as
    these actions are likely incorrect. Filter out these actions for me.
- You need to ensure that each step is at the finest granularity.
- You should strictly follow the output format in the example.

## Here is the task:
### Task
{task}
```

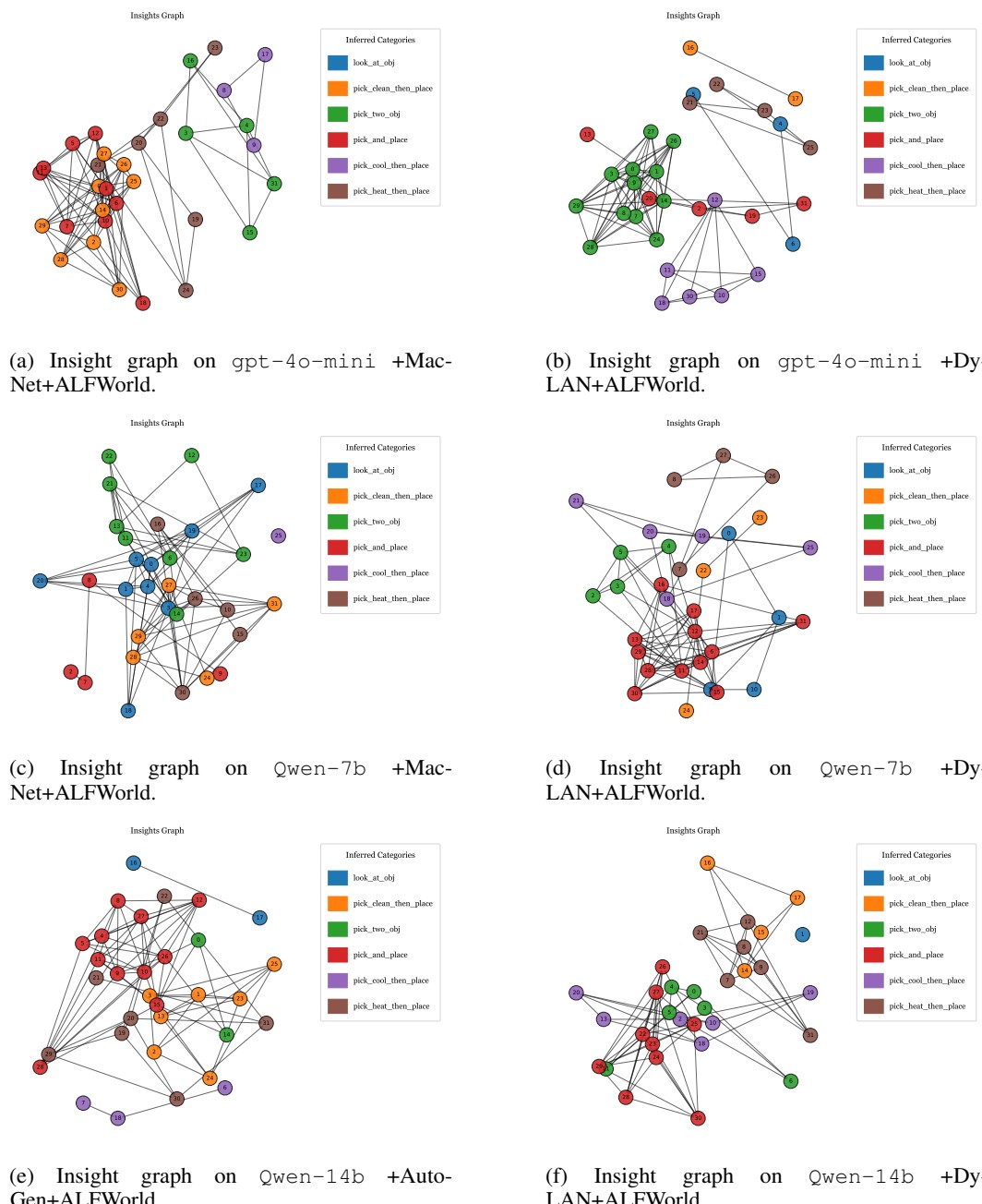

(a) Insight graph on `gpt-4o-mini` +Mac-Net+ALFWorld.

(b) Insight graph on `gpt-4o-mini` +Dy-LAN+ALFWorld.

(c) Insight graph on `Qwen-7b` +Mac-Net+ALFWorld.

(d) Insight graph on `Qwen-7b` +Dy-LAN+ALFWorld.

(e) Insight graph on `Qwen-14b` +Auto-Gen+ALFWorld.

(f) Insight graph on `Qwen-14b` +Dy-LAN+ALFWorld.

Figure 8: Visualizations of insight graphs across different LLM backbones, MAS, and benchmarks.

```
### Trajectory
{trajectory}

### Output
"""
```

The prompt below is partially adapted from [42]. We would like to express our sincere gratitude for their valuable implementation.

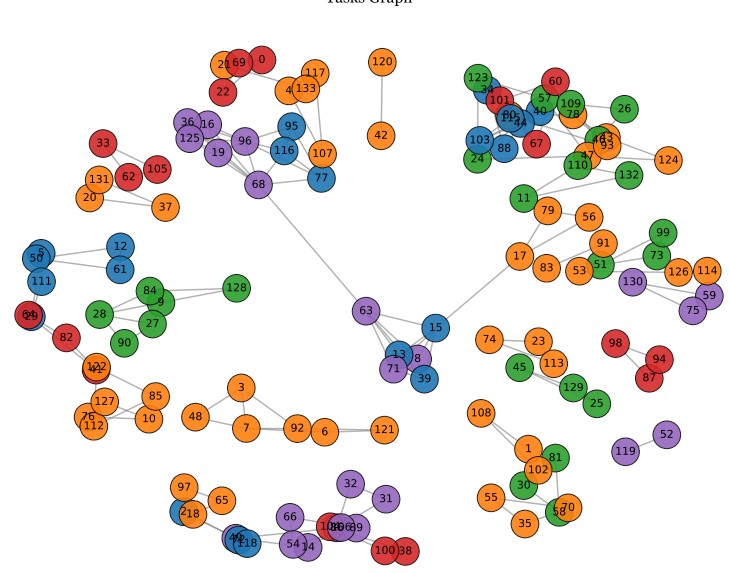

Figure 9: Query graph optimized from ALFWorld dataset.

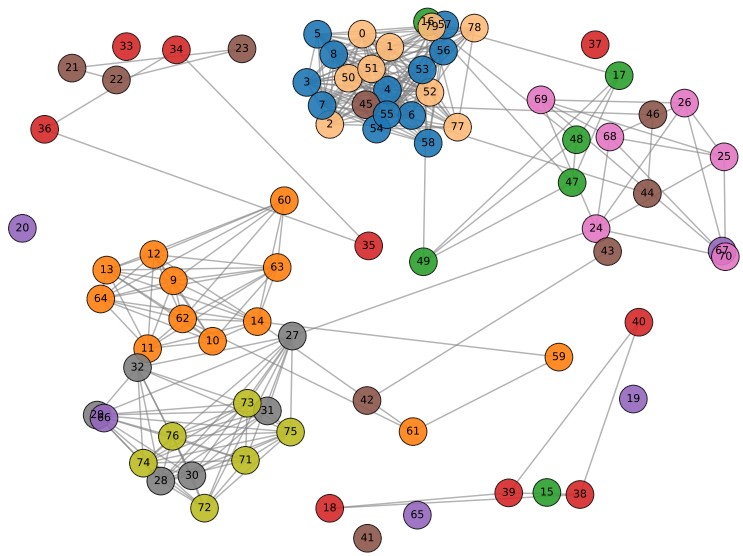

Figure 10: Query graph optimized from SciWorld dataset.

## Inisght Summarization Function

```
learn_lessons_system_prompt_compare = """
You are an analysis-driven agent focused on learning from experience. You will be
    provided with:
- A failed trajectory and its outcome,
- A successful trajectory completing a similar task.

Your task is to analyze both trajectories and generate clear, actionable insights.
     Your insights should highlight what the failed trajectory missed and how the
     successful one addressed or avoided these pitfalls.

## Requirements:
- All insights must be derived directly from contrasting the two trajectories.
```

Tasks Graph

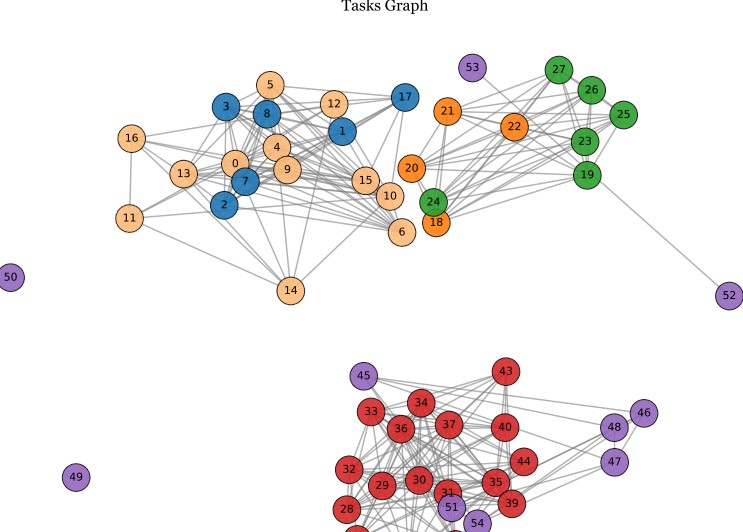

Figure 11: Query graph optimized from PDDL dataset.

```
- Do not speculate or introduce steps not supported by the successful example.
- Focus on **concrete behavioral or strategic differences** between the two cases.
- Keep each insight concise and impactful.

Output Format:
- Start immediately with a numbered list.
- No introduction or explanation.
- Use this exact format:
1. Insight 1
2. Insight 2
3. Insight 3
...
"""

learn_lessons_user_prompt_compare = """
## Successful trajectory
{true_traj}

## Failed trajectory
### trajectory
{false_traj}

Your output:
"""

learn_lessons_system_prompt_all_succ = """
You are an analysis-driven agent focused on learning from success. You will be
    provided with a set of successful trajectories that completed a similar task.

Your goal is to analyze these successful examples and extract clear, actionable
    insights that capture what contributed to their success. These insights will
    serve as guidance for future agents working on similar tasks.

## Requirements:
- All insights must be grounded in patterns or strategies observed across the
    successful trajectories.
- Do not speculate or introduce steps not reflected in the provided examples.
- Focus on common behaviors, strategies, or decisions that consistently led to
    positive outcomes.
- Keep each insight concise, specific, and impactful.

Output Format:
- Start immediately with a numbered list.
- No introduction or explanation.
- Use this exact format:
1. Insight 1
```

```
2. Insight 2
3. Insight 3
...
"""

learn_lessons_user_prompt_all_succ = """
## Successful trajectorys
{true_trajs}

Your output:
"""

# merge rules prompt
merge_rules_system_prompt = """"You are an agent skilled at summarizing and
    distilling insights. You are given a list of insights that were previously
    extracted from similar tasks. These insights may contain redundancy or
    overlap.

Your job is to **merge and consolidate similar insights**, and output a refined
    version that is **clear, actionable, and concise**.

NOTE:
- All merged insights **must be based strictly on the given inputs**. You are **
    not allowed to make up** or infer any new information.
- The output should be easy to read and follow.

Output Format:
- Start your response directly with the numbered list, no preamble or explanations
    .
- Each insight should be a short sentence.
- Use the following format exactly:
1. Insight 1
2. Insight 2
3. Insight 3
...
"""

merge_rules_user_prompt = """
## Here are the current insights that need to be merged:
{current_rules}

## Please consolidate and rewrite them into **no more than {limited_number}
    refined insights**.

As the summarizing agent, remove redundancies, combine similar ideas, and ensure
    clarity.

Your output:
"""
```

## Customizing Memory for Agents

```
project_insights_system_prompt: str = """
You are a thoughtful and context-aware agent. You will be provided with a
    successfully executed trajectory, a specific agent **role**, and a set of **
    general insights** applicable across all roles.
Your task is to **adapt these general insights** into **personalized insights**
    that are specifically tailored to the given role and its trajectory. These
    personalized insights should help the agent improve future performance by
    aligning with their unique background, responsibilities, and perspective.
Make sure your output reflects an understanding of the role's context and promotes
     actionable, role-relevant advice.

NOTE - Your output must strictly follow the format below:
1. Insight 1
2. Insight 2
3. Insight 3
...
"""

project_insights_user_prompt: str = """
### Trajectory
{trajectory}

### Agent's Role:
```

```
{role}

### General Insights:
{insights}

### Your Output (Personalized Insights for This Role):
"""
```

## D   Discussion with Related Works

In this section, we further discuss the relationship between `G-Memory` and several recent agent memory frameworks. For **A-Mem** [60], while both A-Mem and `G-Memory` aim to enhance the memory capabilities of LLM agents, they differ in two key aspects. First, A-Mem is tailored for single-agent scenarios, whereas `G-Memory` is designed for processing MAS's lengthy and nuanced interaction trajectory. Second, A-Mem emphasizes atomic memory construction for chatbot-style interactions, while `G-Memory` focuses on distilling reusable strategies from collaborative task execution, where fine-grained atomicity is neither required nor beneficial. For **Mem0** [61], although it also employs a graph-based structure, it remains within the chatbot paradigm. Its graph is closer to a knowledge graph, where nodes represent factual entities and edges represent relations, fundamentally differing from `G-Memory`'s agent-centric memory graphs that encode trajectories, decisions, and coordination patterns across agents.

