# OpenReview forum: "G-Memory: Tracing Hierarchical Memory for Multi-Agent Systems"
_NeurIPS.cc/2025/Conference — NeurIPS 2025 spotlight_

### Official Review · Reviewer_adrQ · 2025-07-02

**Clarity:** 2
**Significance:** 3
**Originality:** 3
**Rating:** 4
**Confidence:** 2

**Summary:**

This paper introduces G-Memory, a novel hierarchical and agentic memory architecture for Large Language Model (LLM)-driven Multi-Agent Systems (MAS). It aims to fill the gap of sophisticated memory mechanisms in MAS, which are currently limited compared to single-agent systems. G-Memory proposes a three-tier memory hierarchy: Insight Graph, Encodes high-level, generalizable lessons from past collaborative experiences. Query Graph, Stores task-specific meta information and tracks relationships between queries. Interaction Graph, Contains fine-grained, role-specific utterances and interactions for each query/task.Upon a new query, G-Memory performs efficient bi-directional retrieval, providing each agent with high-level insights and relevant historical trajectories tailored to their role. The system updates all memory levels after each trial, supporting self-evolution and improved performance. Empirical results show significant performance gains across various benchmarks, frameworks, and LLMs, without excessive resource overhead. The framework is modular and plug-in, compatible with different LLM-based MAS designs.

**Questions:**

- How does the memory graph scale with increasing agents/queries/tasks?
   - Are there mechanisms for automatic pruning or compression of obsolete information?

**Ethical Concerns:**

["NO or VERY MINOR ethics concerns only"]

**Final Justification:**

Thanks to the author for the reply, I'll keep my score

**Limitations:**

The paper discusses generalizability and suggests future evaluation on more diverse domains.

**Quality:**

3

**Strengths And Weaknesses:**

Strengths:
The authors clearly identify the shortcomings of existing MAS memory systems, contrasting them with the requirements of MAS. The hierarchical design is theoretically motivated, practically justified, and systematically implemented. The cross-agent, multi-level approach with role- and trial-adaptive retrieval is novel.

Weaknesses: The paper admits that tasks are limited to five benchmarks, mainly QA, embodied action, and games. Generality to other domains is unproven. Some lower-level details, such as precise prompt engineering and insight validation, rely on in-LLM processing. The robustness of these steps is not deeply analyzed.

---

> ### Author Rebuttal · Authors · 2025-07-30
>
> We sincerely thank you for your careful comments and thorough understanding of our paper! Here we give point-by-point responses to your comments and describe the revisions we made to address them.
>
>
>
> ---
>
> > **`Weakness 1`: The paper admits that tasks are limited to five benchmarks, mainly QA, embodied action, and games. Generality to other domains is unproven.**
>
> Thank you for your valuable suggestion. To further demonstrate the generalizability and effectiveness of G-Memory across diverse task settings, we add the GAIA benchmark, which includes tasks like **file reading, web browsing, coding, and multimodal understanding**. We integrate G-Memory into the SmolAgent framework and evaluate its **performance, API cost, and latency**:
>
>
> |Level|Metric|Vanilla|Voyager|G-Memory|
> |-|-|-|-|-|
> |Overall|Perf.|39.80|36.71|44.80|
> |    |API Cost|$18.1|$25.8|$24.0|
> |    |Latency|8.7h|10.3h|11.4h|
> |Level 1|Perf.|52.80|50.60|56.60|
> |    |API Cost|$5.7|$7.3|$7.0|
> |    |Latency|1.8h|2.0h|2.7h|
> |Level 2|Perf.|34.60|42.30|44.18|
> |    |API Cost|$8.9|$11.3|$9.9|
> |    |Latency|4.9h|5.6h|6.1h|
> |Level 3|Perf.|16.70|16.70|23.07|
> |    |API Cost|$3.5|$7.2|$7.1|
> |    |Latency|2.0h|2.7h|2.6h|
>
> These results suggest that G-Memory consistently improves agent performance across all task levels, with only marginal latency and API cost increase. We will incorporate this analysis into the revised manuscript to further clarify G-Memory’s scalability and practical utility.
>
>
> ---
>
> > **`Weakness 2`: Some lower-level details, such as precise prompt engineering and insight validation, rely on in-LLM processing. The robustness of these steps is not deeply analyzed.**
>
> Thank you for your insightful question! To address your concern, we elaborate on both the intuitive design and empirical effectiveness of our prompt engineering and insight validation mechanisms.
>
> **(1) prompt settings**
>
> Our work incorporates three main components of prompt engineering, which we detail below:
>
> * **LLM-based Graph Sparsifier (Eq. 7):**
>   The operator $\mathcal{S}_\text{LLM}$ is designed to distill a verbose interaction graph $\mathcal{G}\_\mathsf{inter} = (\mathcal{U}, \mathcal{E}\_\mathsf{u})$ into a minimal yet sufficient reasoning path tailored to a new query $Q$. This is achieved in two stages: **(1) Node Summarization**, where an LLM function condenses the textual content $m\_i$ of each utterance node $u\_i = (\mathcal{A}\_i, m\_i)$, producing a summarized node set $\mathcal{U}'$ while preserving the original graph structure. This yields an abstracted graph $\mathcal{G}\_\text{sum} = (\mathcal{U}', \mathcal{E}\_\mathsf{u})$, where $\mathcal{U}' = \\{ (\mathcal{A}\_i, \text{LLM}\_\text{sum}(m\_i)) \mid (\mathcal{A}\_i, m\_i) \in \mathcal{U} \\}$; **(2) Graph Extraction** Subsequently, the abstracted graph $\mathcal{G}\_\text{sum}$ is then provided to $\mathcal{S}\_\text{LLM}$ and produces $\hat{\mathcal{G}}\_\mathsf{inter}(\hat{\mathcal{U}}, \hat{\mathcal{E}}) = \mathcal{S}\_\text{LLM}(\mathcal{G}\_\text{sum}, Q)$.
>   To further validate the necessity of this component, we conduct an ablation study comparing systems with and without $\mathcal{S}\_\text{LLM}$:
>
> |MAS|Variant|ALFWorld|SciWorld|PDDL|
> |-|-|-|-|-|
> |AutoGen|w/ $\mathcal{S}\_\text{LLM}$|88.81|67.40|27.77|
> ||w/o $\mathcal{S}\_\text{LLM}$|86.36|62.36|24.56|
> |DyLAN|w/ $\mathcal{S}\_\text{LLM}$|70.90|65.64|18.95|
> ||w/o $\mathcal{S}\_\text{LLM}$|68.13|64.16|16.09|
>
> These results empirically confirm that the LLM-based interaction summarization and filtering mechanism is both effective and essential.
>
> * **LLM-based Filtration (Eq. 7):**
>   As referenced in Eq. (7), the filtration process is guided by an LLM, and we provide the full implementation details in Appendix C for transparency and reproducibility.
> * **Insight Extraction (Eq. 10):**
>   This is detailed below:
>
>
>
> **(2) insight validation**
>
> G-Memory is equipped with at least two mechanisms to ensure that accurate and helpful insights are increasingly emphasized over time:
>
> - **G-Memory periodically performs consolidation over accumulated insights to merge redundant ones and discard irrelevant or obsolete content.** First, recent queries are grouped into semantically related clusters $\{\mathcal{C}\_l\}$. For each cluster, all corresponding insights are retrieved from the graph: $\mathcal{I}\_l = \Pi_{\mathcal{Q} \to \mathcal{I}}(\mathcal{C}\_l)$, which is then synthesized into a set of more general insight, $\iota_l^\text{merged}$, using the function $\mathcal{J}$ in Eq. (10) that merges their textual content and supporting query sets: $\iota\_l^\text{merged} = ( \mathcal{J}(\{\kappa\_k \mid (\kappa\_k, \Omega\_k) \in \mathcal{I}\_l\}), \bigcup\_{\iota\_k \in \mathcal{I}\_l} \Omega\_k )$. This consolidation ensures that the insight graph is regularly updated to maintain concise and representative high-level knowledge.
> - As the query and insight graphs evolve over time, correct and helpful insights/interactions naturally lead to more successful task executions, thereby accumulating additional edge connections. This process **positions them as denser and more central nodes in the graph**, making them more likely to be retrieved, as illustrated by nodes 45 and 55 in Figure 10. In contrast, misleading elements tend to have limited impact and remain on the periphery of the graph (e.g., nodes 36 and 37 in Figure 10). This update dynamically implicitly prioritizes reliable knowledge while suppressing noise.
>
> ---
>
> > **`Question 1`: How does the memory graph scale with increasing agents/queries/tasks**
>
> We would like to respectfully point out that all major operations within G-Memory are designed to **scale efficiently** without significantly increasing token cost **as the queries/tasks grow**:
>
>
> |Memory Layer|Formula|Process|LLM invocation|Detail|
> |-|-|-|-|-|
> |Query-level|Eq. (4)|Retrieval|No|based on embedding similarity|
> ||Eq. (9)|Update|No|-|
> |Interaction-level|Eq. (5) and (7)|Retrieval|Yes|The LLM receives $\tilde{\mathcal{Q}}^S$ as input, i.e., the $k$ queries retrieved by Eq. (4) and their 1-hop neighbors. Assuming an average node degree $\overline{D}$, the total input size is approximately $k \cdot \overline{D}$, which does not increase significantly with the number of nodes.|
> ||-|Update|No|
> |Insight-level|Eq. (6)|Retrieval|No
> ||Eq. (10)|Update|Yes|This process takes the new query and its interaction graph as input. The input size remains roughly constant.
>
>
> In addition to being robust to the number of queries/tasks, **G-Memory is also largely insensitive to the number of agents**, as almost all processes and operators within G-Memory are **independent of the agent team size**. The only exception is Eq. (8), where the computational cost scales linearly with the number of agents. Nevertheless, this cost remains manageable in practice and does not significantly affect overall efficiency.
>
> In summary, G-Memory can scale its memory pool without incurring excessive token cost, making it suitable for long-horizon multi-agent tasks.
>
>
>
>
>
> ----
>
> > **`Question 2`: Are there mechanisms for automatic pruning or compression of obsolete information?**
>
>
> Thank you for this insightful inquiry! Please refer to our response to `Weakness 2`, where we introduce the consolidation mechanism where insights are periodically reconstructed or merged.

---

> ### Comment · Area_Chair_qW8p · 2025-08-05
>
> Hi Reviewer adrQ,
>
> Can you check the author response and see if you would like to follow up on any remaining concern?
>
> Thanks,
> AC

---

### Official Review · Reviewer_o31T · 2025-07-02

**Clarity:** 3
**Significance:** 3
**Originality:** 3
**Rating:** 5
**Confidence:** 4

**Summary:**

This paper introduces G-Memory, a hierarchical memory system designed to enhance the self-evolution capabilities of multi-agent systems (MAS) by addressing the limitations of existing memory architectures. G-Memory organizes complex inter-agent interactions into a three-tier graph structure to enable efficient retrieval of high-level insights and fine-grained collaboration trajectories. The system improves task performance on different kinds of tasks. Experiments demonstrate its effectiveness as a plug-and-play module, facilitating adaptive and role-specific memory support for agents.

**Questions:**

See above weaknesses

**Ethical Concerns:**

["NO or VERY MINOR ethics concerns only"]

**Final Justification:**

The authors have addressed my concerns.

**Limitations:**

yes

**Quality:**

3

**Strengths And Weaknesses:**

**Strengths**

1. The writing is clear and engaging. Even though the memory system is complex, the authors explain it in a way that’s easy to follow. The paper flows well, the key points stand out, and the figures help a lot. It’s a solid, well-structured read.

2. The three-tier memory design is intuitive. It doesn’t just store past interactions but organizes them in a way that makes sense. The insight, query, and interaction graphs help agents pull out both high-level lessons and fine-grained details when needed.

3. It improves performance without blowing up costs. A 20%+ boost in some tasks while keeping token usage reasonable is impressive. A lot of memory systems add overhead, but this one seems efficient.

4. The experiments cover all the bases. They test performance, break down costs, run ablation studies, and include case studies. It’s thorough and convincing.

**Weaknesses**

1. What about the cold-start problem? To my understanding, the system needs some initial memory to work well, but the paper doesn’t explain how that’s built or how sensitive performance is to the order of tasks. If early tasks feed bad data into memory, does that mess things up later? Some analysis here should be done.

2. How exactly are token costs calculated? Is it just completion tokens, or does it include prompts? And does the size of memory affect this? The memory pool becomes larger gradually, what's the impact on the cost? At the initial stage, the cost could be few, but the cost might be higher later, though the argtop operators could make the cost insensitive to the total volume of the memory pool. The details should be introduced.

3. Why skip the 0-hop baseline in Figure 4? The hop expansion is supposed to fix noisy similarity matches, but without testing a no-hop version, it’s hard to tell if expanding actually helps or just adds more noise. Also, the motivation for hop expansion in line 173 "the similarity may be only superficial or noisy" seems weird to me. By using hop expansion, more nodes could be considered, and maybe more noise is introduced.

4. The ranking/sparsification part feels handwavy. Eq. 7 mentions it, and the appendix only includes some prompts. How exactly does the LLM decide what to keep or drop? What's the effectivenss? A clearer explanation would make it easier to replicate.

Overall, it’s a good paper to me, but a few more details, especially on initialization and implementation, would make it even better. If authors could address my concerns, I could consider further raise my score.

---

> ### Author Rebuttal · Authors · 2025-07-30
>
> We would like to express our deepest respect for your meticulous review! We can genuinely sense that you have dedicated considerable time to thoroughly reviewing our manuscript and providing very specific insights and feedback. We must acknowledge that this is one of the most enlightening and helpful reviews we have received in recent years! In response to your efforts, we have carefully prepared a point-by-point reply:
>
> ---
>
> > **`Weakness 1`: What about the cold-start problem?**
>
>
> Thank you for this important question! We respectfully detail the cold-start process for each memory layer as follows:
>
> * **Query Graph**: Each node represents a task query, and a new node is added upon the completion of each task. Retrieval in this graph does not involve a cold-start process—each query retrieves similar past ones via Eq. (4). To reduce early-stage noise caused by sparse or weakly correlated data, we apply a hard similarity threshold; when no historical query exceeds this threshold, no retrieval occurs, effectively suppressing spurious early matches.
> * **Insight Graph**: To ensure meaningful insight-level retrieval, we delay G-Memory’s insight-based querying until 20 queries have been processed. Nevertheless, the accumulation of insight-level data begins from the very start. Nevertheless, G-Memory is not sensitive to the choice of cold-start threshold. We present a sensitivity analysis below, showing that varying the minimum delay (i.e., the number of queries before retrieval is enabled) has a limited impact on performance:
>
> *Table A. Sensitivity study of delay queries using GPT-4o-mini.*
> |Delay|5|10|20|30|
> |-|-|-|-|-|
> |ALFWorld + AutoGen|85.07|87.01|88.81|87.50|
> |ALFworld + DyLAN|69.40|70.90|70.90|71.14|
> |FEVER + AutoGen|65.48|65.48|66.24|65.27|
> |FEVER + DyLAN|62.24|63.69|64.22|66.38|
> * **Interaction Graph**: This graph evolves in sync with the query graph. If relevant past queries are retained (i.e., not filtered out due to low similarity), the corresponding sparsified interaction graphs are available for retrieval. If none are deemed relevant, no retrieval is triggered.
>
> Overall, G-Memory is designed to be **robust to cold-start issues** and can efficiently operate in new domains with minimal prior data. We sincerely thank you again for your thoughtful feedback and will include these clarifications in the revised manuscript.
>
>
> ---
>
> > **`Weakness 2`: How exactly are token costs calculated? And does the size of memory affect this?**
>
>
> Please allow us to respectfully respond to the questions one by one:
>
> **(1) How exactly are token costs calculated?**
>  The token cost demonstrated in Figures 3  and 7 is the sum of both prompt token and completion token.
>
> **(2) Does the size of memory affect this? The memory pool becomes larger gradually, what's the impact on the cost?**
>
> We would like to clarify that all major operations within G-Memory are designed to **scale efficiently** without significantly increasing token cost as the memory pool grows:
>
>
> |Memory Layer|Formula|Process|LLM invocation|Detail|
> |-|-|-|-|-|
> |Query-level|Eq. (4)|Retrieval|No|based on embedding similarity|
> ||Eq. (9)|Update|No|-|
> |Interaction-level|Eq. (5) and (7)|Retrieval|Yes|The LLM receives $\tilde{\mathcal{Q}}^S$ as input, i.e., the $k$ queries retrieved by Eq. (4) and their 1-hop neighbors. Assuming an average node degree $\overline{D}$, the total input size is approximately $k \cdot \overline{D}$, which does not increase significantly with the number of nodes.|
> ||-|Update|No|
> |Insight-level|Eq. (6)|Retrieval|No
> ||Eq. (10)|Update|Yes|This process takes the new query and its interaction graph as input. The input size remains roughly constant.
>
>
> In summary, G-Memory can scale its memory pool without incurring excessive token cost, making it suitable for long-horizon multi-agent tasks.
>
>
>
> ---
>
> > **`Weakness 3`: Why skip the 0-hop baseline in Figure 4?**
>
> To address your concern, we have supplemented the sensitivity analysis in Figure 4(a) with additional results for 0-hop:
>
> *Table B. Hop expansion experiment using qwen-2.5-14b.*
> | Benchmark | MAS| 0hop | 1hop | 2hop | 3hop|
> |-|-|-|-|-|-|
> | ALFWorld|AutoGen| 81.34| 85.82 | 79.85 | 79.85 |
> | PDDL|AutoGen |46.58 |55.24 | 49.79 | 56.31 |
> | FEVER |AutoGen | 69.70 |71.43 | 72.45 | 71.71 |
> | ALFWorld| DyLAN | 75.30 |81.34 | 78.36 | 77.61 |
> | PDDL |DyLAN |48.54 |51.12 | 49.58 | 44.94 |
> | FEVER |DyLAN| 65.31 | 66.66 | 63.27 | 64.29 |
>
> These results demonstrate a trade-off between the receptive field and noise as the hop size increases. With fewer hops (e.g., 0-hop), the retrieved queries may be too sparse or uninformative. Conversely, expanding to larger hops may introduce excessive noise. We observe that **1–2 hops strike a reasonable balance**.
>
> Therefore, we agree that “the similarity may be only superficial or noisy” applies both when the hop expansion is too limited or too broad. Thank you again for your insightful comment! We will clarify this point in Section 5.4 of the revised manuscript.
>
> ---
>
> > **`Weakness 4`: The ranking/sparsification part feels handwavy.**
>
> To address your concern, we will humbly (1) provide a more fine-grained explanation of the sparsifier, and (2) empirically validate its effectiveness.
>
>
> **（1）Detailed Process**
>
> The $\mathcal{S}\_\text{LLM}$ operator distills a verbose interaction graph $\mathcal{G}\_\mathsf{inter}=(\mathcal{U}, \mathcal{E}\_\mathsf{u})$ into its essential reasoning path relative to a new query $Q$. This is performed in two main stages: **(1) Node Summarization**, where an LLM function condenses the textual content $m_i$ of each utterance node $u\_i = (\mathcal{A}\_i, m\_i)$, producing a summarized node set $\mathcal{U}'$ while preserving the original graph structure. This yields an abstracted graph $\mathcal{G}\_\text{sum} = (\mathcal{U}', \mathcal{E}_\mathsf{u})$, where $\mathcal{U}' = \\{ (\mathcal{A}\_i, \text{LLM}\_\text{sum}(m\_i)) \mid (\mathcal{A}\_i, m\_i) \in \mathcal{U} \\}$; **(2) Graph Extraction**
> Subsequently, the abstracted graph $\mathcal{G}\_\text{sum}$ is then provided to $\mathcal{S}\_\text{LLM}$ and produces $\hat{\mathcal{G}}\_\mathsf{inter}(\hat{\mathcal{U}}, \hat{\mathcal{E}}) = \mathcal{S}\_\text{LLM}(\mathcal{G}\_\text{sum}, Q)$.
>
> **(2) Empirical Evalution**
>
> We further conducted an ablation study on this component, and the results are presented below:
>
> *Table C. Ablation study of sparsifier. The backbone LLM is gpt-4o-mini.*
> |MAS|Variant|ALFWorld|SciWorld|PDDL|
> |-|-|-|-|-|
> |AutoGen|w/ $\mathcal{S}_\text{LLM}$|88.81|67.40|27.77|
> ||w/o $\mathcal{S}_\text{LLM}$|86.36|62.36|24.56|
> |DyLAN|w/ $\mathcal{S}_\text{LLM}$|70.90|65.64|18.95|
> ||w/o $\mathcal{S}_\text{LLM}$|68.13|64.16|16.09|
>
> These results confirm that, providing interaction memory in a more concise manner is conducive for multi-agent systems to better leverage previous experience.

---

> > ### Comment · Reviewer_o31T · 2025-08-06
> >
> > Thank you for the detailed response and additional experiments. My concerns have been addressed, and I have raised my score.

---

> > > ### Author Response · Authors · 2025-08-06
> > > **Thank you for your stronger support of our work!**
> > >
> > > Thank you for your kind comment and stronger support of our work. We deeply appreciate your reviews and feedback on the cold-start issue and scalability of G-Memory, which have significantly elevated the quality of our work. It is our honor to address your concerns, and we sincerely thank you once again.

---

> ### Comment · Area_Chair_qW8p · 2025-08-05
>
> Hi Reviewer o31T,
>
> Can you check the author response with new results and see if all your questions were addressed?
>
> Thanks,
> AC

---

### Official Review · Reviewer_PY4E · 2025-07-05

**Clarity:** 4
**Significance:** 3
**Originality:** 3
**Rating:** 5
**Confidence:** 3

**Summary:**

This paper introduces G-Memory, a hierarchical memory system for Large Language Model powered multi-agent systems (MAS). The authors identify that existing MAS memory architectures are oversimplified, lacking nuanced inter-agent collaboration tracking and cross-trial customization, hindering self-evolution. G-Memory addresses this via a three-tier graph hierarchy: Insight Graph, Query Graph and interaction graph. Upon a new query, G-Memory performs bidirectional traversal (upward for insights, downward for relevant interaction subgraphs) to provide role-specific memory cues. After task execution, the graphs evolve by assimilating new experiences. Evaluated across five benchmarks and three MAS frameworks, G-Memory improves success rates significantly. The system operates as a plug-and-play module, requiring no framework modifications.

**Questions:**

1. How are the three-tier graphs initialized during the bootstrapping process?
2. Are all new queries incorporated into the memory? If not, what criteria filter queries for updates?
3. How is the correctness of insights verified, and does insight quality impact G-Memory's performance?
4. What safeguards prevent adverse effects from noisy/false insights or action trajectories (e.g., hallucinated lessons)?

**Ethical Concerns:**

["NO or VERY MINOR ethics concerns only"]

**Final Justification:**

After carefully reading the rebuttal, I find that the authors have addressed most of my concerns. I believe the paper's novelty and quality meet the conference's acceptance criteria. Therefore, I rate this paper as 5 (accept).

**Limitations:**

yes

**Paper Formatting Concerns:**

no major formatting issues

**Quality:**

3

**Strengths And Weaknesses:**

Strengths
1. Novel architecture: The three-tier graph hierarchy  innovatively captures multi-granularity memory needs, enabling both high-level reasoning  and fine-grained collaboration recall .
2. Significant Performance Gains: G-Memory achieves state-of-the-art improvements across diverse tasks.
3. Resource Efficiency: Despite its advanced capabilities, G-Memory incurs only marginal token overhead
4. Seamless Integration: As a plug-and-play module, G-Memory integrates into mainstream MAS frameworks without structural modifications, demonstrating practical deployability.

Weaknesses
1. Cold-Start Graph Initialization: The paper lacks details on bootstrapping the three-tier graphs from scratch, which is critical for real-world deployment with no prior collaboration data.
2. Ambiguous Graph Update Mechanisms: The update process for assimilating new experiences requires clarification:
3. Safety and Robustness Gaps: Potential risks from retrieving misleading insights/interactions are underexplored, despite the impact statement noting amplification of incorrect reasoning.

---

> ### Author Rebuttal · Authors · 2025-07-30
>
> We sincerely thank you for the thoughtful and constructive reviews of our manuscript! Based on your questions and recommendations, we give point-by-point responses to your comments and describe the revisions we made to address them.
>
> ---
>
> > **`Weakness & Question 1`: Cold-Start Graph Initialization**
>
>
> Thank you for this important question! We respectfully detail the cold-start process for each memory layer as follows:
>
> * **Query Graph**: Each node represents a task query, and a new node is added upon the completion of each task. Retrieval in this graph does not involve a cold-start process—each query retrieves similar past ones via Eq. (4). To reduce early-stage noise caused by sparse or weakly correlated data, we apply a hard similarity threshold; when no historical query exceeds this threshold, no retrieval occurs, effectively suppressing spurious early matches.
> * **Insight Graph**: To ensure meaningful insight-level retrieval, we delay G-Memory’s insight-based querying until 20 queries have been processed. Nevertheless, the accumulation of insight-level data begins from the very start.
> * **Interaction Graph**: This graph evolves in sync with the query graph. If relevant past queries are retained (i.e., not filtered out due to low similarity), the corresponding sparsified interaction graphs are available for retrieval. If none are deemed relevant, no retrieval is triggered.
>
> Overall, G-Memory is designed to be **robust to cold-start issues** and can efficiently operate in new domains with minimal prior data. We sincerely thank you again for your thoughtful feedback and will include these clarifications in the revised manuscript.
>
>
> ---
>
> > **`Weakness 2`: Ambiguous Graph Update Mechanisms: The update process for assimilating new experiences requires clarification**
>
>
> To further address your concern, we provide a more intuitive and detailed explanation of how each layer of the graph is updated:
>
> * **Query Graph**: As described on Line 214, each encountered query is directly added as a node in the query graph. Additionally, previous queries that contributed to solving the current one (e.g., by providing helpful insights) are linked via directed edges. This construction captures not only semantic similarity but also *temporal and causal relationships*, representing how past queries inspire future ones.
> * **Insight Graph**: Newly distilled insights from each incoming query are added to the insight graph. Edges are drawn from previously leveraged insights that contributed to solving the current query. Compared to query-level connections, this high-level topology provides a more precise mapping of transferable collaborative knowledge, enabling MAS to quickly locate relevant prior experience. G-Memory also performs periodic consolidation of the insight graph: First, recent queries are grouped into semantically related clusters $\{\mathcal{C}\_l\}$. For each cluster, all corresponding insights are retrieved from the graph: $\mathcal{I}\_l = \Pi\_{\mathcal{Q} \to \mathcal{I}}(\mathcal{C}\_l)$, which is then synthesized into a set of more general insight, $\iota_l^\text{merged}$, using the function $\mathcal{J}$ in Eq. (10) that merges their textual content and supporting query sets: $\iota\_l^\text{merged} = ( \mathcal{J}(\{\kappa\_k \mid (\kappa\_k, \Omega\_k) \in \mathcal{I}\_l\}), \bigcup\_{\iota\_k \in \mathcal{I}\_l} \Omega\_k )$.
> Finally, the insight graph is atomically updated by replacing the original set of insights $\mathcal{I}\_l$ with $\iota\_l^\text{merged}$. This is explicitly implemented in the `merge_insight` function of `/mas/memory/mas_memory/GMemory.py` in our provided codebase. This iterative refinement prevents knowledge fragmentation and ensures the long-term coherence of the system's memory.
> * **Interaction Graph**: The interaction history associated with each query is stored in the interaction graph for future reference.
>
>
>
>
> ---
>
> > **`Weakness 3 & Question 4`:
> Safety and Robustness Gaps: Potential risks from retrieving misleading insights/interactions are underexplored, despite the impact statement noting amplification of incorrect reasoning.**
>
>
> Thank you for your insightful observation! We would like to respectfully highlight that the current design of G-Memory includes **at least two mechanisms to mitigate the potential negative impact of misleading insights/interactions**:
>
> * The consolidation process of insight graph will filter the useless insights and emphasize the impactful ones, as described in our response to  `Weakness 2`.
> * As the query and insight graphs evolve over time, correct and helpful insights/interactions naturally lead to more successful task executions, thereby accumulating additional edge connections. This process positions them as denser and more central nodes in the graph, making them more likely to be retrieved, as illustrated by nodes 45 and 55 in Figure 10. In contrast, misleading elements tend to have limited impact and remain on the periphery of the graph (e.g., nodes 36 and 37 in Figure 10). This update dynamic implicitly prioritizes reliable knowledge while suppressing noise.
>
>
>
>
> ---
>
> > **`Question 2`: Are all new queries incorporated into the memory? If not, what criteria filter queries for updates?**
>
>
> Both the **query** and its associated **interaction** history are incorporated into the memory. However, the **insights** distilled from these interactions may be dynamically merged or pruned, as described in our response to  `Weakness 2`.
>
>
> ---
>
> > **`Question 3`:
> How is the correctness of insights verified, and does insight quality impact G-Memory's performance?**
>
> Thank you for your thoughtful question! We agree that the quality of insights can indeed influence the performance of G-Memory. While G-Memory does not explicitly evaluate the correctness of each insight (which is often impractical, as even human-generated insights may appear correct in the short term but prove inaccurate over time), our graph update mechanism implicitly addresses this issue. Specifically, helpful and positively contributing insights naturally accumulate more edge connections and are thus positioned as **denser and more central nodes** in the graph, while misleading ones are gradually marginalized. Additionally, the consolidation process of the insight graph further prunes or merges low-quality or redundant insights.

---

> ### Comment · Area_Chair_qW8p · 2025-08-05
>
> Hi Reviewer PY4E,
>
> Can you please check if the author response has addressed your questions and provided the details you were looking for?
>
> Thank you!
> AC

---

### Official Review · Reviewer_Lxdg · 2025-07-05

**Clarity:** 3
**Significance:** 4
**Originality:** 4
**Rating:** 5
**Confidence:** 4

**Summary:**

In this paper, the authors propose a three-layer graph memory, including insight graph, query graph, and interaction graph. The proposed G-memory enables agents to store, recall, and reason over past experiences more effectively.

**Questions:**

question 1: see weakness 1

question 2: How about the performance of G-memory in more domains, such as on WebArena dataset or GAIA dataset?

question 3: How about the latency of the G-memory; In the paper, the authors present the token cost, but the latency is not mentioned.

**Ethical Concerns:**

["NO or VERY MINOR ethics concerns only"]

**Final Justification:**

In this paper, the authors propose a novel three-layer graph memory. During the rebuttal, they clarify previously missing concepts, which has increased my confidence in my score. My main concern was the performance of the proposed memory system across different tasks. The authors addressed this by providing additional experiments that demonstrate its effectiveness across various tasks. Regarding my second concern about latency, the authors balance performance gains with inference overhead, showing that G-Memory outperforms existing methods. Therefore, I am raising both my score and my confidence level.

**Quality:**

4

**Strengths And Weaknesses:**

Strengths:

1. The design of the three layers hierarchical memory framework is novel. It captures the high-level insights, intermediate queries, and interaction between agents.

2. The authors conduct experiment across multiple domains, including knowledge reasoning, embodied action, game. Experimental results demonstrate the effectiveness of the proposed method.

3. The authors also conduct a series of ablation study to validate the performance.

Weakness:

1.  Some notations are not very clear. Such as:

      a. What is one epoch in MAS? In debate case, one round of agent speaking is one epoch; but in solely task solving case, what does one epoch mean?

      b. In equation (5), $\mathcal{N}^{+}$ is? Is it out-neighbor?

2. In the experiment section, it is expected to adopt more tasks to validate the effectiveness of the proposed method.

---

> ### Author Rebuttal · Authors · 2025-07-30
>
> Thank you immensely for your time and efforts, as well as the helpful and constructive feedback! Here, we give point-by-point responses to your comments.
>
>
>
> ---
>
> > `Weakness & Question 1`: Some notations are not very clear. What is one epoch in MAS? In debate case, one round of agent speaking is one epoch; but in solely task solving case, what does one epoch mean?
>
> Thank you for pointing out this issue. We would like to respectfully clarify the following:
>
> * In general, an *epoch* refers to a single iteration of the agent interaction process, typically defined by one full cycle of decision making or action execution within the multi-agent workflow. More specifically:
>
>   * In frameworks such as **LLM-debate** and **DyLAN**, an epoch corresponds to one complete round of agent responses, as you rightly noted.
>   * In coding frameworks, such as  **MetaGPT** and **EvoMac**, an epoch denotes one round of code generation, which often concludes with a reviewer or tester evaluating the output and feeding back signals for the next round.
>   * In **AutoGen**, where a single round of interaction typically produces a final action, the number of epochs is naturally set to 1.
>
> * In Eq. (5), $\mathcal{N}^+(Q_j)$ and $\mathcal{N}^-(Q_j)$ denote the out-neighborhood and in-neighborhood of node $Q_j$, respectively.
>
> We greatly appreciate your careful reading and will incorporate these clarifications into the revised manuscript.
>
>
> ---
>
> > `Weakness & Question 2`: In the experiment section, it is expected to adopt more tasks to validate the effectiveness of the proposed method; ... such as on WebArena dataset or GAIA dataset?
>
> Thank you for your valuable suggestion. To further demonstrate the generalizability and effectiveness of G-Memory across diverse task settings, we integrate G-Memory into the SmolAgent framework and evaluate its **performance, API cost, and latency** on the GAIA benchmark across multiple levels:
>
>
> |Level|Metric|Vanilla|Voyager|G-Memory|
> |-|-|-|-|-|
> |Overall|Perf.|39.80|36.71|44.80|
> |    |API Cost|$18.1|$25.8|$24.0|
> |    |Latency|8.7h|10.3h|11.4h|
> |Level 1|Perf.|52.80|50.60|56.60|
> |    |API Cost|$5.7|$7.3|$7.0|
> |    |Latency|1.8h|2.0h|2.1h|
> |Level 2|Perf.|34.60|42.30|44.18|
> |    |API Cost|$8.9|$11.3|$9.9|
> |    |Latency|4.9h|5.6h|5.7h|
> |Level 3|Perf.|16.70|16.70|23.07|
> |    |API Cost|$3.5|$7.2|$7.1|
> |    |Latency|2.0h|2.7h|3.6h|
>
> These results suggest that G-Memory consistently improves agent performance across all task levels, with only marginal increases in latency and API cost. We will supplement the experiment into the revised manuscript to further clarify G-Memory’s scalability and practical utility.
>
>
>
> ---
>
> > `Question 3`: How about the latency of the G-memory; In the paper, the authors present the token cost, but the latency is not mentioned.
>
> To address your concern, we provide additional latency measurements below:
>
> |MAS|Memory|ALFWorld||SciWorld||
> |-|-|-|-|-|-|
> |||Peformance|Latency (s)|Performance|Latency (s)|
> |AutoGen|No-Memory|77.61|19204|54.59|16953|
> ||Voyager|85.07|21754|62.36|16650|
> ||MemoryBank|74.96|15492|53.11|10104|
> ||Generative|86.36|20682|61.19|16674|
> ||MetaGPT-M|81.34|16021|61.91|15853|
> ||ChatDev-M|79.85|22347|50.96|12904|
> ||MacNet-M|76.55|21089|55.44|16882|
> ||**G-Memory (Ours)**|**88.81**|21113|**67.40**|17326|
> |DyLAN|No-Memory|56.72|33520|55.38|32408|
> ||Voyager|66.42|29628|62.83|31633|
> ||MemoryBank|55.22|31813|54.74|32925|
> ||Generative|67.91|31010|64.16|34038|
> ||MetaGPT-M|69.40|22936|62.37|32049|
> ||ChatDev-M|46.27|29739|53.35|33111|
> ||MacNet-M|53.44|24991|54.32|34815|
> ||**G-Memory (Ours)**|**70.90**|26726|**65.64**|33447|
>
>  As shown, (1) **G-Memory incurs only moderate inference overhead even when delivering the most significant performance gains**. For instance, on AutoGen+ALFWorld, it increases latency by merely 9% compared to the no-memory baseline; (2) **in some cases, G-Memory even improves efficiency**, e.g., on DyLAN+ALFWorld, it leads to a 20% speed-up, as helpful memory cues enable the multi-agent system to reach correct actions more quickly and terminate the interaction earlier.

---

> > ### Comment · Reviewer_Lxdg · 2025-08-05
> >
> > Thank you for the feedback and thorough experiments. The additional results have fully addressed my concerns, and I am pleased to raise my score accordingly.

---

> > > ### Author Response · Authors · 2025-08-06
> > > **Thank you for your stronger support of our work!**
> > >
> > > We would like to sincerely thank you for your inspiring discussion and stronger support of our work. We are more than happy to see that our rebuttal has properly addressed your concerns!

---

### Decision · Program_Chairs · 2025-09-17

**Decision:**

Accept (spotlight)

**Comment:**

Summary:

This paper introduces G-Memory, a hierarchical, agentic memory system for multi-agent systems (MAS) inspired by organizational memory theory. It manages the lengthy MAS interaction via a three-tier graph hierarchy: insight, query, and interaction graphs. The paper conducts extensive experiments with three LLM backbones and three MAS frameworks on multiple benchmarks, and shows the performance gains brought by G-Memory without changing the original MAS framework.

Strengths:

1. The proposed memory system is novel. Reviewers all appreciate the design and clear distinction from existing memory systems.

2. The experiments are comprehensive by showing performance and cost and conducting ablation studies and case studies. They look thorough and convincing. The memory system can also be seamlessly integrated into multiple MAS frameworks as a plug-and-play module.

3. The paper is very well-written with a clear structure and carefully curated figures/tables.

Weaknesses:

Before the rebuttal, reviewers asked many clarification questions, including notations and definitions of epochs, graph initialization, and ambiguous graph update mechanism, to which the authors have provided answers during the rebuttal.

On the experiment side, during the rebuttal period, the authors added new results on the GAIA benchmark, conducted more analysis on the latency of G-memory, sensitivity, and the scaling efficiency as the queries/tasks grow.

Overall, all reviewers are satisfied with the responses and some have increased their scores. I would highly recommend accepting the paper, and also encourage the committee to consider a spotlight/oral award.